# Transcriptomic comparison of two selective retinal cell ablation paradigms in zebrafish reveals shared and cell-specific regenerative responses

Kevin Emmerich[1,2,�], Steven L. Walker[3,�,¤a], Guohua Wang[1,�,¤b], David T. White[1], Anneliese Ceisel[1], Fang Wang[1], Yong Teng[4], Zeeshaan Chunawala[1], Gianna Graziano[1], Saumya Nimmagadda[1], Meera T. Saxena[1], Jiang Qian[1], Jeff S. Mumm[1,2,3,5]*

1 Department of Ophthalmology, Wilmer Eye Institute, Johns Hopkins University, Baltimore, Maryland, United States of America, 2 McKusick-Nathans Institute of the Department of Genetic Medicine, Johns Hopkins School of Medicine, Baltimore, Maryland, United States of America, 3 Department of Cellular Biology and Anatomy, Medical College of Georgia, Augusta University, Augusta, Georgia, United States of America, 4 Department of Hematology and Medical Oncology, Winship Cancer Institute, Emory University, Atlanta, Georgia, United States of America, 5 Solomon H. Snyder Department of Neuroscience, Johns Hopkins University, Baltimore, Maryland, United States of America

These authors contributed equally to this work.
¤a Current address: School of Biomedical Sciences, The Chinese University of Hong Kong, N.T., Hong Kong
¤b Current address: School of Computer Science and Technology, Harbin Institute of Technology, Harbin, People's Republic of China
* jmumm3@jhmi.edu

**Data Availability Statement:** Raw microarray expression data that support the findings of this study have been deposited in GEO with the

## Abstract

Retinal Müller glia (MG) can act as stem-like cells to generate new neurons in both zebrafish and mice. In zebrafish, retinal regeneration is innate and robust, resulting in the replacement of lost neurons and restoration of visual function. In mice, exogenous stimulation of MG is required to reveal a dormant and, to date, limited regenerative capacity. Zebrafish studies have been key in revealing factors that promote regenerative responses in the mammalian eye. Increased understanding of how the regenerative potential of MG is regulated in zebrafish may therefore aid efforts to promote retinal repair therapeutically. Developmental signaling pathways are known to coordinate regeneration following widespread retinal cell loss. In contrast, less is known about how regeneration is regulated in the context of retinal degenerative disease, i.e., following the loss of specific retinal cell types. To address this knowledge gap, we compared transcriptomic responses underlying regeneration following targeted loss of rod photoreceptors or bipolar cells. In total, 2,531 differentially expressed genes (DEGs) were identified, with the majority being paradigm specific, including during early MG activation phases, suggesting the nature of the injury/cell loss informs the regenerative process from initiation onward. For example, early modulation of Notch signaling was implicated in the rod but not bipolar cell ablation paradigm and components of JAK/STAT signaling were implicated in both paradigms. To examine candidate gene roles in rod cell regeneration, including several immune-related factors, CRISPR/Cas9 was used to create G0 mutant larvae (i.e., "crispants"). Rod cell regeneration was inhibited in *stat3* crispants, while mutating *stat5a/b*, *c7b* and *txn* accelerated rod regeneration kinetics. These data

accession code GSE234646. Additionally, processed microarray data directly supporting the results in this manuscript (pattern analyses, pathway results and DEGs) can be found in S3–S8 Tables.

**Funding:** This project was funded by the following grant awards: F31-EY032790 from NIH/NEI (KE); T32-EY7143-22 from NIH/NEI (KE); F31-EY021713 from NIH/NEI (SLW); R01-EY022810 from NIH/NEI (JSM); R01-EY033009 from NIH/NEI (JSM); Vision Discovery Institute – Pilot Project from the Medical College of Georgia (JSM); 5-FY10-7 Basil O'Connor Starter Scholar Research Award from the March of Dimes (JSM). Salary support from funders was received by KE (NIH/NEI), SLW (NIH/NEI), and JSM (NIH/NEI March of Dimes). The funders had no role in study design, data collection and analysis, decision to publish, or preparation of the manuscript.

**Competing interests:** I have read the journal's policy and the authors of this manuscript have the following competing interests: JSM holds patents for the NTR inducible cell ablation system (US #7,514,595) and uses thereof (US #8,071,838 and US#8431768).

support emerging evidence that discrete responses follow from selective retinal cell loss and that the immune system plays a key role in regulating "fate-biased" regenerative processes.

## Author summary

Blinding diseases are linked to the loss of specific types of neurons in the retina. In humans, this eventually leads to loss of sight. In zebrafish, however, lost retinal neurons are regenerated resulting in restored vision. Our lab has developed zebrafish models that induce the loss of disease-relevant retinal neurons, thereby allowing us to study how individual cell types are regenerated. Here, to better understand how these processes are regulated, we compared gene expression changes occurring during loss and regeneration of two different retinal cell types, rod photoreceptors and bipolar interneurons. The majority of gene changes were specific to each cell type studied, providing strong evidence that genetic programs underlying stem cell activation vary depending on the cell type lost. We also found that the immune system was implicated as a regulator of regeneration in both models, but that individual immune-related genes were more strongly associated with one of the two models. Furthermore, disrupting multiple genes involved in immune system signaling led to enhanced rod regeneration. We hope that a better understanding of how retinal cell regeneration is regulated in zebrafish will aid efforts to develop regenerative therapeutics designed to restore sight to patients who have lost their vision.

## Introduction

Regenerative therapeutics aimed at replacing lost retinal cells have the potential to restore visual function. One category of therapeutics in development aims to stimulate endogenous retinal Müller glia (MG) to act as stem-like cells to replace lost neurons [1,2]. The capacity of MG to drive retinal repair differs markedly between species. In mammals, MG regenerative potential is intact but remains dormant unless exogenously stimulated. In contrast, zebrafish MG robustly regenerate lost retinal neurons [3]. Zebrafish are therefore an established model for exploring mechanisms controlling MG regenerative potential, including factors that enhance the regenerative potential of mammalian MG [4–8]. Studies of the mechanisms regulating regeneration following widespread retinal cell loss, e.g., damage across multiple cell types/nuclear layers, have focused largely on the role of known developmental pathways and reprogramming factors [3,9–12]. In contrast, mechanisms governing responses to the loss of specific neuronal types are less well defined. Intriguingly, recent studies in zebrafish suggest that the loss of discrete retinal cell types elicits a fate-biased regenerative response; i.e., where progenitors cells give rise predominantly to the lost cell type [13–16]. A deeper understanding of the mechanisms controlling fate-biased regeneration has the potential to facilitate the development of novel cell-type targeted regenerative therapeutic strategies.

Retinal regenerative research in fish has utilized three main injury paradigms: light-induced ablation [17–20], toxin-induced ablation [21,22] and mechanical wounding [23,24]. Early mechanical wounding studies involved surgical excisions of retinal quadrants which induce classic hallmarks of epimorphic regeneration: rapid wound closure and the formation of blastema-like zones of proliferative cells [23]. Other paradigms eliminate all or most cells in a retinal somal layer, e.g., light-induced ablation of rod and cone photoreceptor cells in the outer

nuclear layer (ONL) [17–20], excitotoxin-induced ablation of interneurons in the inner nuclear layer (INL) and retinal ganglion cells (RGCs) [21,22]. Collectively, these studies have identified two injury-inducible stem cell niches in the teleost retina; ONL precursors committed to producing rod cells [25,26], and an INL-localized retinal stem cell [27,28]. Subsequent studies demonstrated that the INL stem cells were Müller glia (MG), which function as injury-induced multipotent stem-like cells. These cells can divide asymmetrically giving rise to a transit-amplifying pool of retinal progenitor cells that then differentiate to regenerate the lost retinal cells [3,9–11]. Intriguingly, mammalian MG can also produce new neurons upon exogenous expression of neurogenic factors, such as *achaete-scute family bHLH transcription factor 1* (*ascl1*) [4,5] or inhibition of factors such as *nuclear factor I a/b/x* (*nfia/b/x*), which function to sustain MG identity [29]. However, the types of neurons produced by mammalian MG are not typically well correlated to types of cells lost; for instance, *ascl1* overexpression is largely limited to the production of bipolar cells [5]. More recently, combinatorial application of exogenous neurogenic factors have overcome this limitation to produce ganglion-like cells capable of responding to light [7]. These studies highlight the need to better understand the regulation of MG regenerative potential, and fate choices of MG-derived neural progenitors, are regulated.

Retinal degenerative diseases typically involve the loss of discrete neural cell types. Prominent examples of this include rod photoreceptor loss in retinitis pigmentosa and RGC loss in glaucoma [30,31]. Thus, methods for replacing specific cell types are being sought as a strategy to restore vison to patients. To study how discrete cell types are regenerated we adapted a targeted cell ablation method, the nitroreductase (NTR) prodrug converting enzyme system [32–36], to zebrafish [37–42]. Transgenic expression of NTR renders cells susceptible to prodrugs, such as metronidazole (Mtz). Exposing fish to Mtz results in the selective loss of NTR-expressing cells by a DNA-damage induced cell death process [41–44]. Co-expression of a fluorescent reporter in NTR-targeted cells allows the regenerative process to be characterized in detail using *in vivo* time-lapse imaging [39,45] or quantified using high-throughput screening methods [40,46]. This approach affords key advantages to the investigation of cellular regeneration. For instance, it can be genetically targeted to individual cell types and is prodrug inducible, allowing temporal synchronization of the regenerative process across many samples. While several mutant zebrafish lines exhibit cell-type specific losses, the lack of control over the onset and duration of cell death make it more difficult to profile regenerative responses in these models. Additionally, several retinal degeneration mutants also fail to regenerate [47] suggesting that a threshold of cell death is required to trigger retinal regeneration. Using this approach, recent studies have shown that selective retinal cell loss elicits a fate-biased regenerative response [13–16], although subtler fate-biases have also recently been acknowledged in widespread injury paradigms as well. This suggests limited/selective retinal cell loss and widespread/non-selective retinal injury may trigger regenerative processes that are differentially regulated.

Here, to increase understanding of fate-biased regenerative processes, we used time-resolved gene expression profiling to compare two retinal cell-type specific ablation paradigms in zebrafish. Comparisons between the targeted loss of either 1) bipolar cells [45], or 2) rod photoreceptors [46] involved profiling gene expression changes across 12 timepoints spanning the entire degenerative and regenerative processes. Differential gene expression analysis across a total of 138 microarrays revealed both shared and paradigm-specific profiles. Paradigm-specific expression changes predominated at early and late time points, and shared expression profiles peaked at 24 and 32 hours after induction of cell loss. While paradigm-specific mechanisms were expected to be prevalent during late phases that correspond to progenitor cell differentiation, predominance at early time points suggests that initial retinal stem cell activation

involves signals that may be informative regarding cell type lost, thus facilitating fate-biased responses. Finally, genetic disruption of multiple immune-related genes was found to inhibit or enhance rod regeneration kinetics in crispant larvae. This information regarding fate-biased regeneration and immune system modulation could enable the development of cell-type specific regenerative therapeutics for the mammalian retina.

# Results

## Specificity of NTR/Mtz-induced cell death and assessments of regeneration

To expand our understanding of mechanisms regulating the regeneration of discrete retinal cell types, gene expression changes attending two cell-specific ablation paradigms were compared. Specifically, transgenic zebrafish lines expressing NTR-reporter fusions in either a bipolar interneuron subtype (*nyctalopin* or *nyx*-expressing) or rod photoreceptor cells were used, referred to hereafter as "NTR-bipolar" [45] and "NTR-rod" [46], respectively (see Methods for full description). We first confirmed that NTR expression was specific to the targeted cell types within the retina. Prior studies have shown that expression in the NTR-bipolar line is restricted to retinal bipolar cells, with labeling of the pineal gland (epiphysis) in the brain as well [45, 48]. For the NTR-rod line, immunohistochemistry confirmed that expression of the NTR-YFP fusion protein was restricted to rod cells, as previously shown for this line [41] and corresponding promoter element [49]. NTR-YFP expressing cells were co-labeled with a rod-specific antibody marking outer segments, zpr-3 [50, 51](S1a–S1a" Fig), and interdigitated with cells marked by a cone-specific antibody, zpr-1 [52] (S1b–S1b" Fig). Co-labeling levels with zpr-3 suggested that the majority of rod cells express the NTR-YFP transgene. The NTR-bipolar line uses the Gal4/UAS bipartite system to express two different transgenes, a membrane-tagged YFP and an NTR-mCherry fusion protein that enables cell ablation. In this background, NTR-mCherry expression is sporadic, with the number of NTR+ cells varying from low (~25 cells/retina) to high (hundreds of cells/retina) [45]. To account for this variability, mid-level expressing NTR-bipolar larvae (~100 targeted cells) were visually selected for these studies.

We next verified that a 24 hr 10 mM Mtz prodrug treatment was sufficient to induce loss of NTR-expressing cells and that regeneration occurs upon Mtz washout. Confocal time series imaging was used to follow this process in individual NTR-bipolar and NTR-rod larva (Figs 1a–1b series and S2b and S2d series, respectively). After a 24 hr 10 mM Mtz treatment (from 6–7 days post-fertilization, dpf) followed by 24 hr washout, NTR+ cells in each paradigm were largely absent or appear fragmented (Fig 1a' and 1b') in comparison to pre-treatment images (Fig 1a and 1b), indicating strong ablation was achieved. Four days after Mtz was removed (11 dpf), both NTR+ bipolar and rod cells had regenerated (Fig 1a" and 1b"). DMSO controls showed sustained NTR expression in the absence of Mtz (S2a and S2c Fig series). Terminal deoxynucleotidyl transferase (TdT)-mediated dUTP nick end labeling (TUNEL) [53] staining confirmed layer-specific cell death in both paradigms (Figs 1c–1f series and S2e and S2f series). Quantification showed a statistically significant increase in DNA damage (TUNEL staining) following Mtz exposures for both transgenic lines (Fig 1g).

## Characterization of retinal stem cell niches responding to bipolar and rod cell death

As the NTR-rod line consistently labels the majority of rod photoreceptor cells (S1a Fig series), we anticipated rod cell loss would trigger proliferation in both the INL stem cell niche and in ONL precursors, consistent with data from the Hyde lab using a similar transgenic line [54]. Conversely, we expected bipolar cell ablation would result in proliferation predominantly

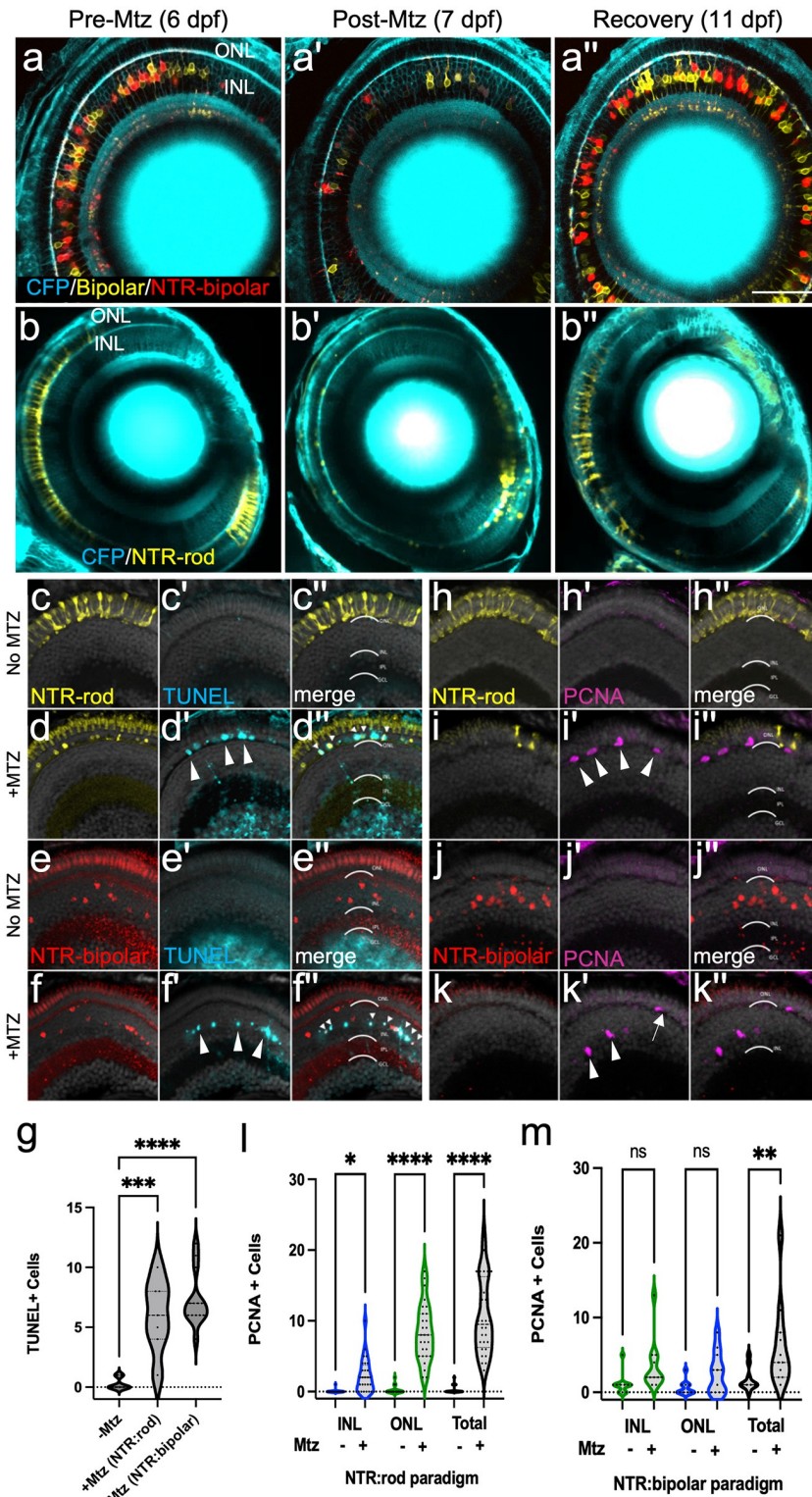

**Fig 1. Two Nitroreductase (NTR) models enabling inducible retinal cell-type specific ablation.** (a-b series) In vivo time-series imaging following the NTR-bipolar (a-a") or NTR-rod (b-b") response to metronidazole (Mtz) treatment to induce cell death and regeneration. Images were taken in the same fish at 6 dpf (before Mtz onset, a, b), 7 dpf (after treatment, a', b') and 11 dpf (following recovery, a", b"). Larvae for each ablation paradigm express CFP derived from Tg(pax6-DF4:gap43-CFP)q01 to label general retinal structures. (c-f series) Representative histological staining for

TUNEL (terminal deoxynucleotidyl transferase dUTP nick end labeling) at 7 dpf in uninjured NTR-rod larvae (c-c") and Mtz treated NTR-rod larvae (d-d") as well as in uninjured and Mtz-treated NTR-bipolar larvae (e-e", f-f', respectively). (g) Quantification of TUNEL+ cell counts at 7 dpf in untreated fish (-Mtz) or following the ablation of each cell type [+Mtz (NTR-rod) and +Mtz (NTR-bipolar)]. For the statistical analysis, Welch's one-way ANOVA was followed by student's t test with Dunnett's method for multiple comparisons correction. (h-k series). Representative histological staining for PCNA (proliferative cell nuclear antigen) at 9 dpf in uninjured NTR-rod larvae (h-h") and Mtz treated NTR-rod larvae (i-i") as well as in uninjured (j-j") and Mtz-treated NTR-bipolar larvae (k-k"). (l,m) Quantification of PCNA+ cell counts at 9 dpf in (l) untreated NTR-rod fish or following the ablation of rods (m) and in untreated NTR-bipolar fish or following the ablation of bipolar cells. For both paradigms, quantification is split into total number of proliferative cells and those in the INL or ONL. For statistical comparisons, Student's t test was used to assess the indicated paired conditions. Asterisks indicate the following p-value ranges: * = $p<0.05$, ** = $p<0.01$, *** = $p<0.001$, and **** = $p<0.0001$, "ns" indicates $p>0.05$.

within the INL. INL proliferation is indicative of activation of MG to stem-like state and expansion of a pool of MG-derived progenitor cells (MGPCs). Proliferation of isolated ONL cells is indicative of activation of fate-restricted rod precursor cells. To test which retinal stem cell niches were responsive to cell loss in the two ablation cell paradigms, Mtz-treated and DMSO control retinas were immunolabeled with proliferative cell nuclear antigen (PCNA [39]) at 72h post-onset of Mtz ablation (Figs 1h–1k series and S2g and S2h series). The results showed rod loss led to increased proliferation in both the ONL and INL (Figs 1i', arrowheads, and 1l and S2h' arrowheads). In contrast, bipolar loss only led to increased proliferation when combining INL and ONL counts (Figs 1k', arrowheads, and 1m and S2g' arrowheads). Both ablation paradigms produced a similar proliferative response in the INL (an average of 2.2 ±2.3, and 3.5 ±3.4 PCNA+ cells for the rod and bipolar paradigms, respectively; student's t test p-value = 0.25; compare Fig 1l to 1m). Importantly, this suggests that MG/MGPC responses were comparable in rod and bipolar cell ablated larvae. In contrast, and as expected, ONL proliferation was higher in the rod versus bipolar paradigm (an average of 8.3 ±4.0, and 2.6 ±2.7 PCNA+ cells for the rod and bipolar paradigms, respectively; student's t test p-value < 0.001). Note, proliferation within the INL in NTR-rod lines was typically clustered where the density of cell loss was high (S2h' Fig), consistent with increased rod cell death activating MG/MGPC proliferation [54]. Note, cells dividing in the ONL following bipolar cell ablation (S2g' Fig, arrowheads) were presumed to be MGPCs that had migrated to the ONL, consistent with prior studies [3].

## Microarray analysis of two cell-specific regeneration paradigms

We next turned to a differential gene expression comparison of our two retinal neuron ablation paradigms. To account for expression changes spanning cell loss and regeneration, eyes were collected from NTR-bipolar and NTR-rod larvae treated ±Mtz across a total of 12 time points, from 0 to 240 hours post ablation (hpa; Fig 2a). As above, cell death was induced by the addition of 10 mM Mtz from 6 to 7 dpf (0 to 24 hpa), followed by wash out and recovery from days 7–16 dpf (24 to 240 hpa; Fig 2a). To assess initial MG stem cell responses to retinal cell loss, early time points were emphasized by collecting samples every 8 hr from 0 to 48 hpa. Eight-hour intervals were used to minimize representation of gene expression changes due solely to circadian cycles. For the later time points, samples were collected at 60, 72, 96, 144, and 240 hpa. Across three biological repeats per time point, a total of 96 eyes were collected per time point for each paradigm (48 DMSO controls and 48 +Mtz samples, S1 Table).

## Genetic differences observed between two cell-specific regeneration paradigms

Using a 1.5-fold change and $\leq 0.05$ *p*-value cutoff compared to expression values at 0 hpa, a total of 2,531 differentially expressed genes (DEGs) were observed across both paradigms.

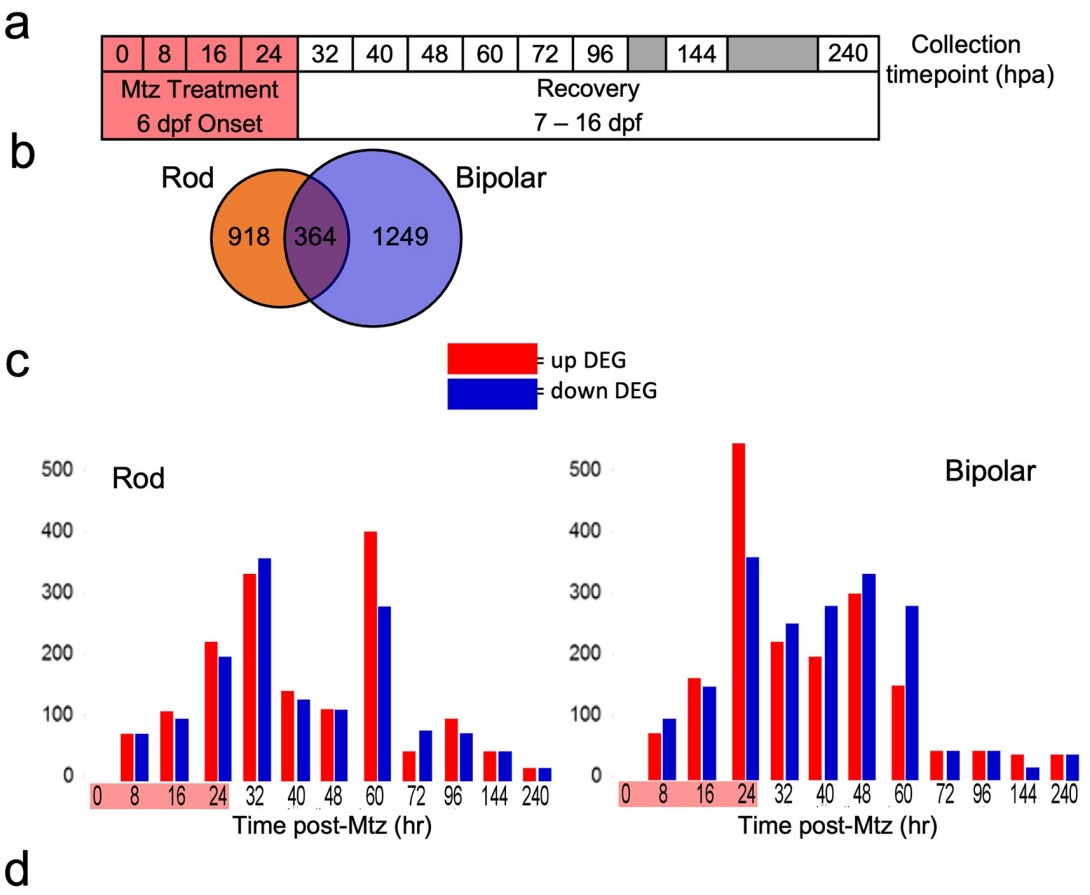

**Fig 2. Microarray data collection, DEG identification and top hits.** (a) Experimental design for tissue collection of whole eyes for microarray analysis. Treatment with Mtz for 24h was induced at 6 dpf following screening for NTR-rod+ or NTR-bipolar+ fish. Eyes were collected in triplicate at the following 12 timepoints including t0, t8, t16, 24, t32, t40, t48, t60, t72, t92, t144 and t240. (b) Venn diagram illustrating the total number of differentially expressed genes (DEGs) unique to either cell type or shared between the two at all timepoints. (c) Chart showing distribution of upregulated and downregulated DEGs in each paradigm. (d) Top 10 up and

downregulated genes across the entire data set based on p-value as well as the number of timepoints that gene was identified as differentially expressed out of 12.

Across all timepoints 918 and 1,249 DEGs were unique to the rod or bipolar cell ablation paradigm, respectively, and 364 DEGs were shared between the two (Fig 2b). DEGs included selective downregulation of known marker genes in the rod (*rho*) and bipolar paradigms (*capb5b*) [29], confirming cell-specific loss in each paradigm. Plotting the number of DEGs per each time point showed two peaks at 32 and 60 hpa for the NTR-rod line, and a single peak at 24 hrs post ablation (hpa) for the NTR-bipolar line (Fig 2c). Next, quantitative reverse transcriptase PCR (qRT-PCR) was used to validate gene expression changes for 8 genes per each paradigm at 0, 24, 48, 72 and 96 hpa (S3 Fig, primers used are listed in S2 Table). Genes in the Janus kinase-signal transducer and activator of transcription (JAK/STAT) and Wnt pathways were selected for qRT-PCR analysis as both have been implicated in regeneration of the targeted cell types [10,55]. Temporally-resolved expression trends, rather than absolute fold change, were utilized for validation as differences in normalization methods have the potential to impact overall expression levels [56]. These data revealed only minor differences in expression patterns between the microarray and qRT-PCR suggesting the microarray data accurately reflect gene expression changes (S3 Fig). The top 10 up and down DEGs at all timepoints for each paradigm were then identified by lowest single *p*-value, many passing the DEG threshold at multiple timepoints during out study (Fig 2d).

## Transcriptomic differences following regeneration of Uniquely targeted neurons

To identify differentially regulated signaling pathways at each time point, and across broader temporally-resolved expression patterns, several analysis methods were used. First, volcano plots and pathfindR [57] analyses were performed to reveal paradigm-specific and shared DEG-associated signaling pathways at each time point (S4–S14 Figs). Volcano plots correlate log-fold changes in gene expression to *p*-values, with upregulated and downregulated DEGs indicated by orange and blue dots, respectively (S4a–S14a Figs). Venn diagrams show the total number of DEGs unique to each paradigm and the number of shared DEGs (S4b–S14b Figs). Kyoto Encyclopedia of Genes and Genomes (KEGG) analysis via the pathfindR package revealed the top enriched pathways associated with either the rod or bipolar paradigm-specific DEGs, as well as shared DEGs (S4c–S14c Figs).

In order to examine DEGs contributing to different stages of the regeneration process, we identified DEGs whose expression profile fit one of four gaussian-type patterns during the regenerative process: increased (up early) or decreased (down early) from16 to 32 hpa, and increased (up late) or decreased (down late) from 48 to 72 hpa (Fig 3a). Notably, this captured the majority of DEGs observed. Heatmaps are shown for the top 25 DEGs for each pattern in both the NTR-rod and NTR-bipolar paradigms (Fig 3b and 3c). The two reciprocally regulated DEG patterns–up/down early (Fig 4) and up/down late (Fig 5)–were evaluated using KEGG, Gene Ontology (GO) and Ingenuity Pathway Analysis (IPA). For early pattern DEGS (Fig 4a), the top 10 pathfindR KEGG terms were plotted for rod-specific, bipolar-specific and shared gene sets (Fig 4b–4d). Predictably for the rod paradigm, phototransduction was the most enriched term. Additional rod-specific KEGG pathways included RNA degradation, proteosome, and necroptosis (Fig 4b). For the bipolar paradigm, p53 signaling was the most enriched term, other pathways included autophagy, nucleoside excision repair, and cysteine and methionine metabolism (Fig 4c). Shared KEGG pathway terms for early pattern DEGs included

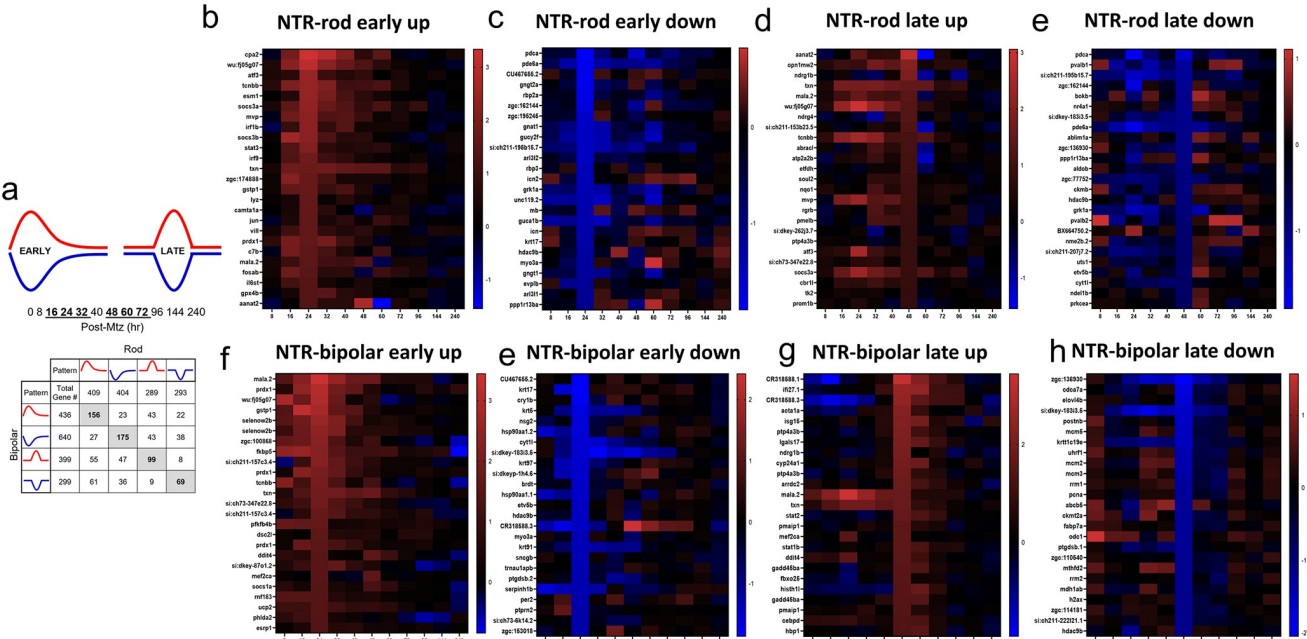

**Fig 3. Identification of early and late DEG patterns and heatmaps for top DEGs in each.** (a) DEGs for each paradigm were split into upregulated (red) and downregulated (blue) as well as early (16-32h) and late (48-72h) based on peaks in Fig 2. The distribution of DEGs in these 8 patterns is shown in terms of unique and shared DEGs across both cell types. (b-h) The top 25 genes are plotted in heatmaps as a function of the largest fold changes at the middle timepoint in each pattern (24h and 60h) to demonstrate top genes defining each 8 patterns (NTR-rod and NTR-bipolar for patterns early-up, early-down, late-up and late-down).

phototransduction, mTor signaling, and glutathione metabolism (Fig 4d). Gene Ontology (GO) analysis of the same datasets revealed additional terms, such as circadian rhythm and ncRNA processing for the rod paradigm, cellular apoptotic processes and response to stress for the bipolar paradigm, and response to temperature stimulus and DNA conformation change among shared terms (Fig 4e–4g). Lastly, IPA was used to generate unsupervised networks from the early DEG lists for the NTR-rod and NTR-bipolar paradigms (Fig 4h and 4i). For the rod paradigm, signal transducer and activator of transcription 3 (STAT3) was identified as the central mediator of the early DEG network along with multiple cytokine terms (e.g, interleukin-1b and 6, IL-1B, IL-6) and associated terms for the oxidative stress response (Fig 4h). In the bipolar paradigm, the central mediator was peroxisome proliferator-activated receptor gamma (PPARG), other noteworthy genes included tumor protein 53 (*tp53*), the cytokine *il10*, *stat5a* and *stat5b* (Fig 4i). We then produced a summary bar chart of all related immune terms identified from IPA. Early pattern IPA-implicated pathways of the rod paradigm included JAK family kinases, chemokine (C-X-C motif) receptor 4 (CXCR4), IL-6, interferon, and iNOS signaling, as well as multiple other IL-family cytokines (S15 Fig). For the bipolar paradigm, early pattern IPA terms included similar IL-family cytokines but also added prolactin and leukocyte extravasation signaling, as well as phagosome maturation (S16 Fig). Combined, KEGG, GO, and IPA analysis revealed that factors implicated in regulating the early stages of regeneration are linked more to the loss of individual retinal cell types than to a common signature across cell types.

The same analysis methods were used to evaluate late pattern DEGs (Fig 5a). A subset of top KEGG terms stayed consistent with early pattern DEGs including phototransduction and RNA degradation for the rod paradigm, autophagy and p53 for the bipolar paradigm, and also

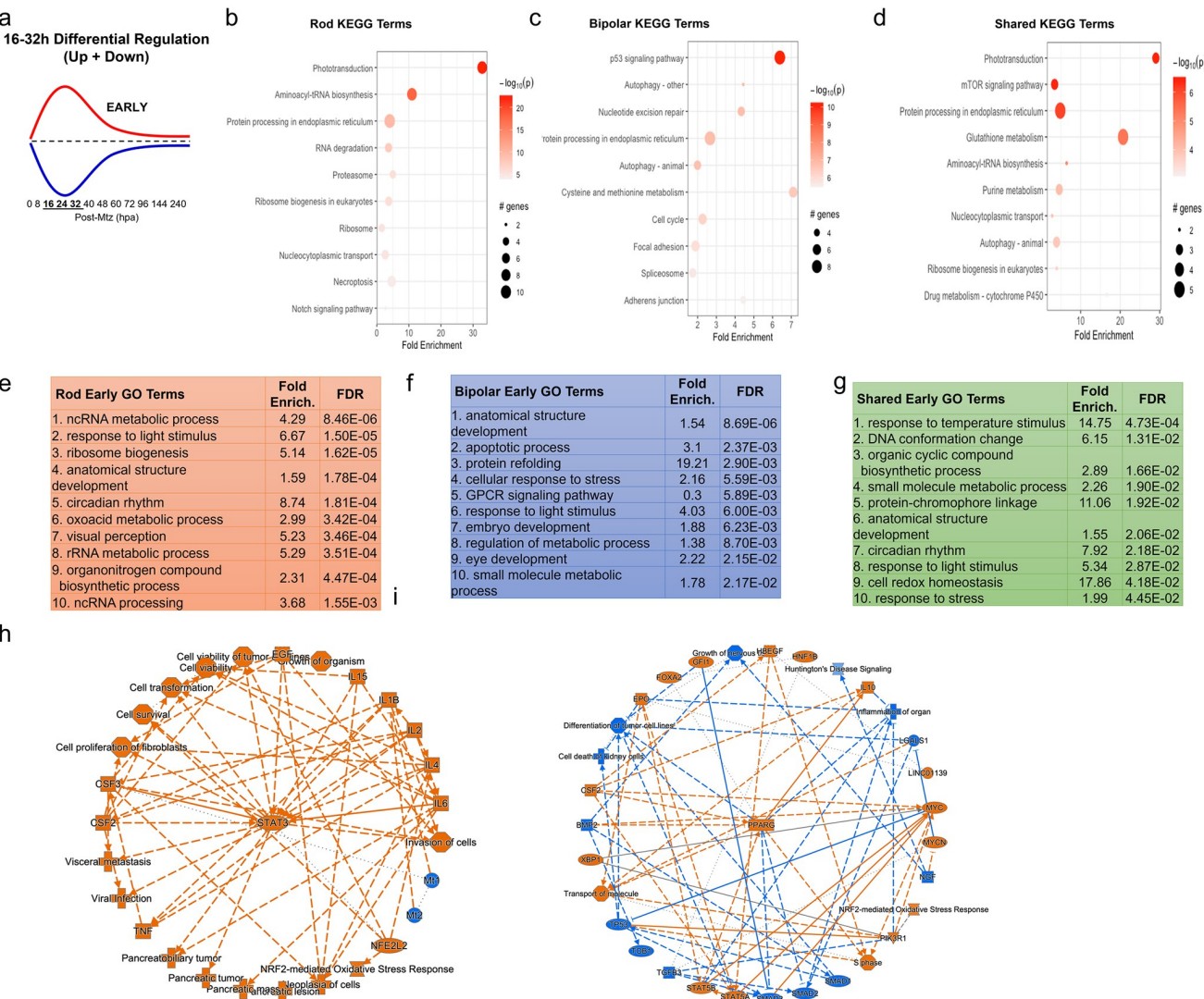

**Fig 4. Pathway analysis of early DEG pattern in both paradigms.** (a) Curve demonstrating expression pattern for early upregulated (red) and downregulated (blue) genes along our timepoints. (b-d) Top 10 enriched KEGG pathway terms including p-value and # of genes found as DEGs in the NTR-rod (b), NTR-bipolar (c) or from shared DEGs across both paradigms (d). (e-g) Top 10 enriched Gene Ontology pathway terms including fold enrichment and false discovery rate in the NTR-rod (e), NTR-bipolar (f) or from shared DEGs across both paradigms (g). (h-i) Unsupervised gene networks produced from ingenuity pathway analysis for DEGs in the NTR-rod (h) and NTR-bipolar (i) paradigm. Orange indicates upregulated gene/pathway term while blue indicates downregulated.

autophagy for shared genes (Fig 5b–5d). The top enriched KEGG terms were DNA replication and cell cycle for the rod and bipolar paradigms, respectively. Newly identified terms included mismatch repair for rods, Toll-like receptor signaling for bipolars, and cellular senescence for the shared set (Fig 5b–5d). Enriched GO terms similarly included some overlap with early pattern terms (Fig 5e–5g), new rod GO terms included ribosome biogenesis, nuclear RNA surveillance, and sno(s) RNA processing (Fig 5e). New bipolar terms included oxoacid metabolic process, circadian rhythm, and response to peptide hormone (Fig 5f). Fewer shared DEGs in the late pattern data lead to fewer enriched GO terms, however, the majority were new including cellular response to DNA damage and regeneration (Fig 5g). IPA analysis of late pattern rod data similarly identified many of the same immune-related pathways implicated in early pattern

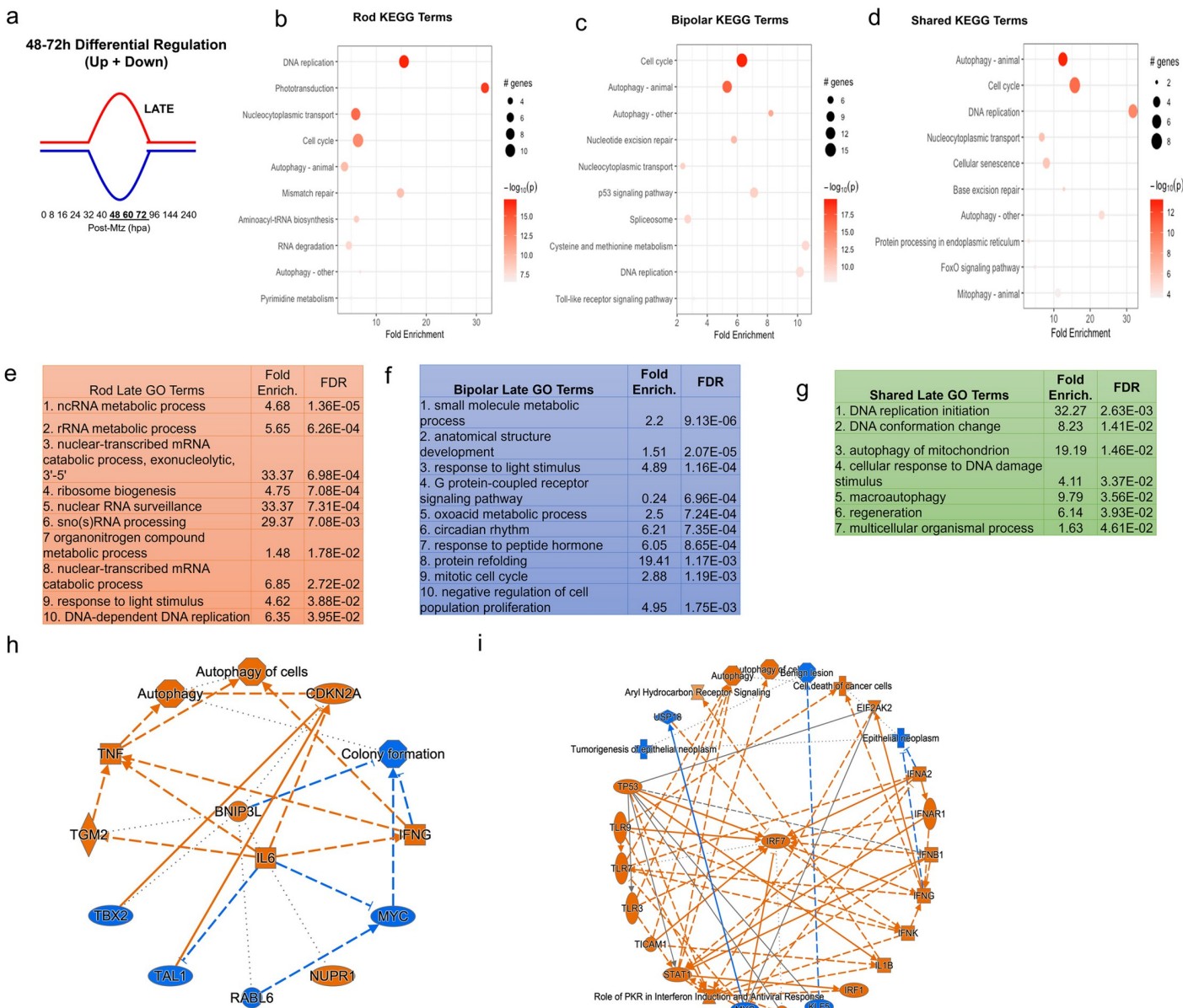

**Fig 5. Pathway analysis of late DEG pattern in both paradigms.** (a) Graphic demonstrating expression pattern for late upregulated (red) and downregulated (blue) genes along our timepoints. (b-d) Top 10 enriched KEGG pathway terms including p-value and # of genes found as DEGs in the NTR-rod (b), NTR-bipolar (c) or from shared DEGs across both paradigms (d). (e-g) Top 10 enriched Gene Ontology pathway terms including fold enrichment and false discovery rate in the NTR-rod (e), NTR-bipolar (f) or from shared DEGs across both paradigms (g, only 7 terms enriched in this group). (h-i) Unsupervised gene networks produced from ingenuity pathway analysis for DEGs in the NTR-rod (h) and NTR-bipolar (i) paradigm. Orange indicates upregulated gene/pathway term while blue indicates downregulated.

rod data, including IL-family cytokines. New terms included prolactin, CNTF, and GM-CSF signaling (S17 Fig). The late pattern rod network had fewer nodes than the early network and was centered on IL-6 (Fig 5h). Late pattern bipolar IPA terms also shared many immune-related terms with early pattern data. Additional terms included CD40, granzyme B, and TNFR signaling (S18 Fig). The late pattern bipolar network was centered on interferon regulatory factor 7 (IRF7, Fig 5i). Finally, a deep literature search was conducted on the top ~50–60 DEGs in each coordinated pattern for each paradigm. More specifically, given the importance of the

immune system in regulating regeneration in our NTR-rod model [39,58], we asked whether these genes had previously been linked to immune responses or regeneration in general (S19a Fig). In total, we found that ~38% and ~44% of the top DEGs in the rod and bipolar paradigms, respectively, had been linked to immune system function. In addition, ~27% of the DEGs from both paradigms had been linked to regeneration in prior studies (S19b Fig).

### Validation of role for *stat3* in induced rod photoreceptor regeneration

The expression of *stat3* increased following retinal cell loss in both the NTR-rod and NTR-bipolar paradigms. However, the increase in *stat3* was far more prominent in the NTR-rod fish paradigm regarding both fold change (S3a Fig) and the number of timepoints this was observed (8 of 12 for NTR-rod, 4 of 12 for NTR-bipolar). For functional tests in NTR-rod fish, CRISPR/Cas9 redundant targeting was used to mutate *stat3* in G0 larvae, i.e., to create crispants [59] (Fig 6a). Reduced expression was confirmed via qPCR in a subset of injected and non-injected control fish (Fig 6b). At 5 dpf, rod photoreceptors were ablated (10 mM 24h Mtz). Subsequently, an established Tecan fluorescent plate reader-based assay as well as *in vivo* imaging was used to assess: rod cell death at 7 dpf (Fig 6c) and rod cell regeneration at 9 dpf (Fig 6d–6f). Mutation of *stat3* led to significantly impaired rod regeneration (-23%) compared to wildtype fish, thereby validating the requirement for *stat3* in rod cell regeneration. Lastly, to examine the mechanism by which *stat3* disruption leads to reduced regeneration, stem-cell proliferation was assessed following ablation in wildtype and mutant fish. Akin to our initial classification of Mtz-induced proliferation (Fig 1l), death of rods led to proliferation in both the INL and ONL. PCNA immunostaining in *stat3* crispants showed a significant decrease in total proliferation compared to wildtype fish (Fig 6g–6i). This suggests *stat3* loss impairs progenitor cell proliferation at this timepoint, leading to fewer regenerated rod cells.

### Enhanced rod photoreceptor regeneration in *txn*, *stat5a/b* and *c7b* crispants

Finally, to identify genes whose disruption could lead to enhanced rod cell regeneration, we tested a panel of 6 additional genes using the crispant strategy. Targets were selected from: immune-related genes upregulated at multiple timepoints following rod ablation (*c7b*, *txn*), genes upregulated during bipolar but not rod regeneration (*pparg*, *stat5a/b*), genes upregulated in MG during photoreceptor regeneration in adult zebrafish (*prdm1a* [12]), or genes implicated in maintaining MG quiescence in mice following a NMDA-induced retinal injury (*nfia* [29]). Sibling NTR-rod larvae were injected with 4xsgRNAs for each gene of interest and rod cells ablated at 5 dpf (10 mM 24h Mtz). Regeneration was then assessed by plate reader at 9 dpf (Fig 7a). Of the six factors tested (Fig 7b and 7c), mutation of three led to enhanced rod regeneration (from least to greatest effect: *txn* (+70%), *stat5a/b* +85%), and *c7b* (+96%), while three had no statistically significant effect (*nfia*, *pparg*, *prdm1a*). Analysis of effects on rod cell development at 5 dpf showed a mild increase following *txn* mutation (+29%), while there was no effect for *stat5a/b* or *c7b* mutations. Lastly, mutation of *txn*, *stat5a/b*, and *c7b* had no effects on the efficacy of Mtz-induced rod cell ablation with YFP levels dropping to non-mutagenized ablated control levels at 7 dpf in all cases (S20 Fig).

### Discussion

Promotion of retinal regeneration would provide a transformative therapeutic approach for patients with blinding degenerative diseases. One strategy to achieve this goal is to induce endogenous human MG to respond to cell loss and regenerate lost retinal neurons, a process that occurs naturally in zebrafish. Zebrafish studies have identified several genes that regulate

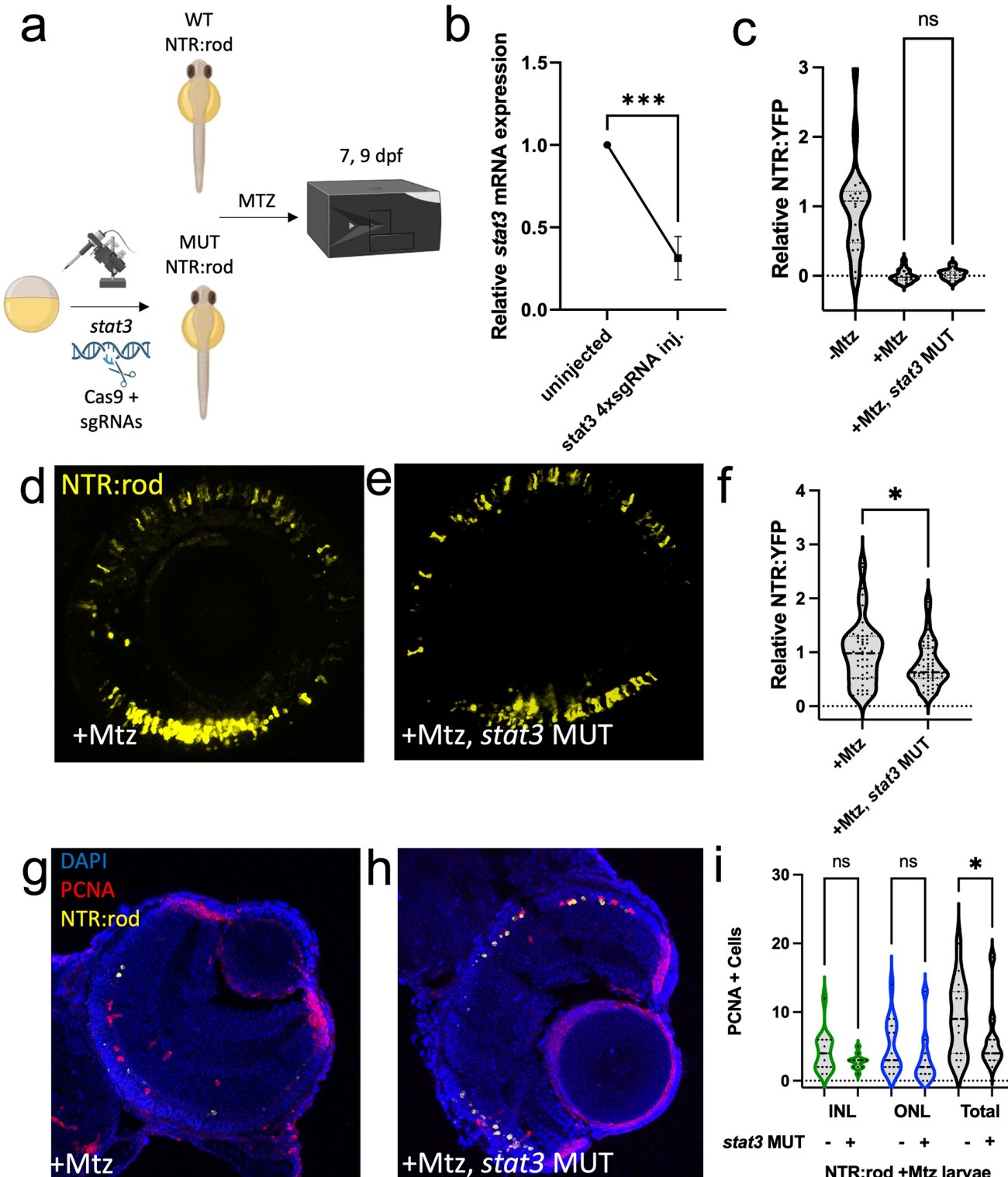

**Fig 6. Stat3 is required for rod photoreceptor regeneration.** (a) Experimental design depicting CRISPR/Cas9-based mutation (MUT) of *stat3* in NTR-rod embryos. Control (+Mtz) and crispant NTR-rod larvae (+Mtz, *stat3* MUT) were treated with 10mM Mtz from 5–6 dpf and rod fluorescence was assessed via plate reader assay at 7 dpf (to examine the extend of rod cell loss) and 9 dpf (to examine the extent of rod cell regeneration). Created with Biorender.com. (b) Relative *stat3* mRNA expression (qRT-PCR assay) in non-injected controls compared to *stat3* crispant fish. (c) Quantification of NTR-YFP expression in rod cells by plate reader assay at 7 dpf in non-ablated (-Mtz) wildtype, and ablated control (+Mtz) and *stat3* MUT larvae (+Mtz,

*stat3* MUT). (d-e) Representative *in vivo* confocal images of regenerated rod cells at 9 dpf in control (+Mtz) and *stat3* MUT larvae (+Mtz, *stat3* MUT). (f) Quantification of rod cell regeneration by plate reader assay at 9 dpf in control (+Mtz) and *stat3* MUT larvae (+Mtz, *stat3* MUT). (g-h) Representative immunohistological staining for PCNA at 7 dpf in control (+Mtz) and *stat3* MUT larvae (+Mtz, *stat3* MUT) fish. (i) Quantification of INL, ONL, and total PCNA+ cells at 7 dpf in control (+Mtz) and *stat3* MUT larvae (+Mtz, *stat3* MUT). For statistical comparisons, Student's t test was used to assess the indicated paired conditions. Asterisks indicate the following p-value ranges: * = p<0.05, ** = p<0.01, *** = p<0.001, and **** = p<0.0001, "ns" indicates p>0.05.

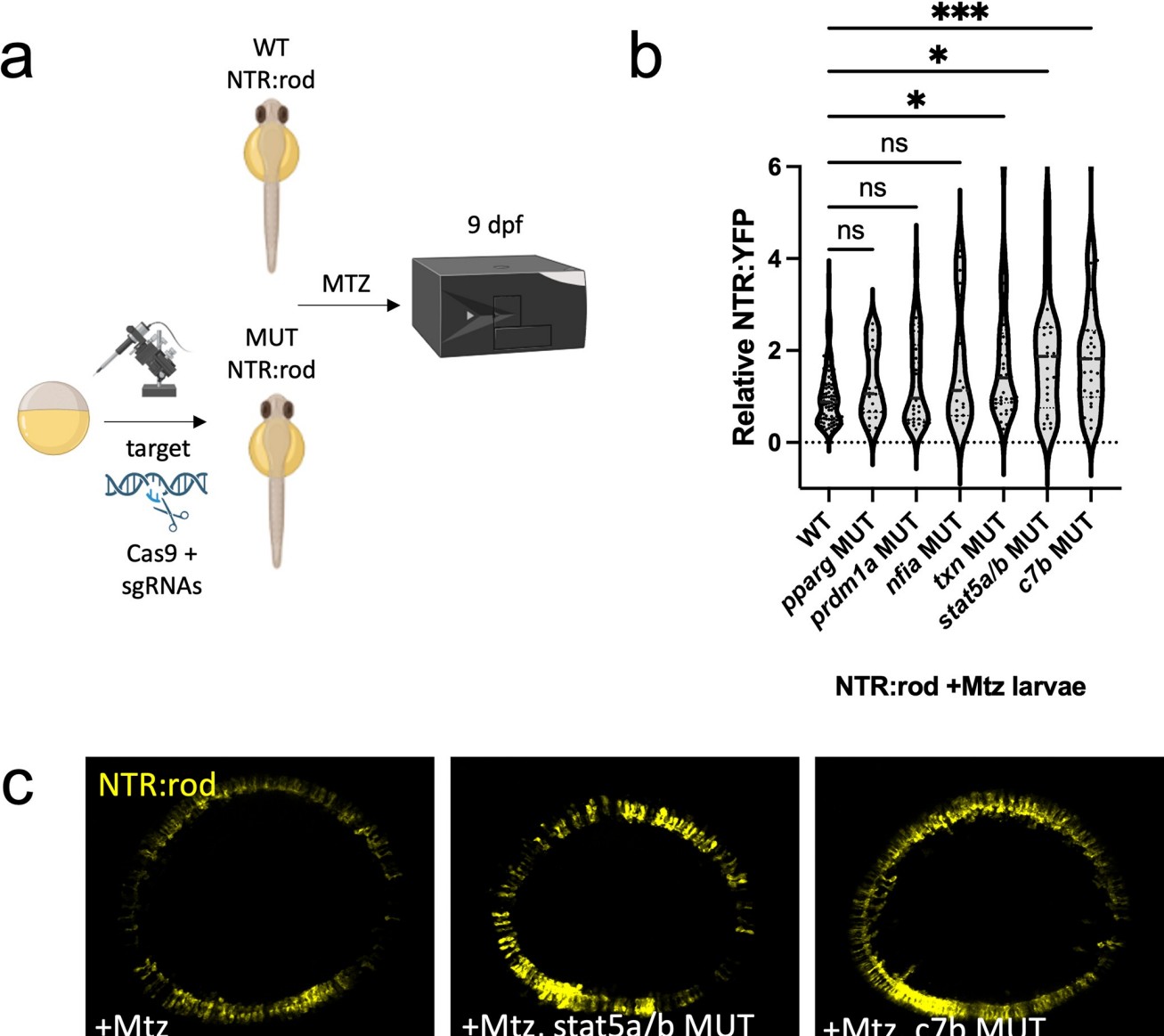

**Fig 7. Rod cell regeneration is enhanced in *txn*, *stat5a/b* and *c7b* crispants.** (a) Experimental design depicting CRISPR/Cas9-based targeting of target genes via CRISPR/Cas9 in NTR-rod embryos. Control (+Mtz) and mutated NTR-rod larvae (+Mtz, *gene target* MUT) were treated with 10mM Mtz from 5–6 dpf and rod fluorescence was assessed via plate reader assay at 9 dpf (to examine the extent of rod cell regeneration). Created with Biorender.com. (b) Quantification of rod cell regeneration by plate reader assay at 9 dpf in Mtz-treated control and MUT larvae for target genes *pparg*, *prdm1a*, *nfia*, *txn*, *stat5a/b* and *c7b*. (c) Representative *in vivo* confocal images of regenerated rod cells at 9 dpf in control (+Mtz) and *stat5a/b* or *c7b* MUT larvae. For statistical comparisons, Welch's one-way ANOVA was followed by student's t test with Dunnett's method for multiple comparisons correction. Asterisks indicate the following p-value ranges: * = p<0.05, ** = p<0.01, *** = p<0.001, and **** = p<0.0001, "ns" indicates p>0.05.

retinal regeneration (*ascl1a* [60], *lin-28 homolog A* [8,11], *sry-box transcription factor 2* [61], *stat3* [62]) as well as gene networks coordinating MG states (rest, reactivity, proliferation) following retinal damage [29]. Each of these studies induced widespread retinal damage where the regenerative process largely mirrors retinal development [3,12]. Conversely, retinal degenerative diseases typically involve the slow progressive loss or dysfunction of specific cell types, such as rod photoreceptors in retinitis pigmentosa [31] and retinal ganglion cells in glaucoma [30,31]. Interestingly, several zebrafish mutants fail to regenerate retinal neurons when loss is slow, progressive, and cell-type specific, i.e., akin to disease conditions [47]. This suggests that a threshold of cell death is required to trigger retinal regeneration and that models enabling control over the onset, extent, specificity and duration of cell death can expand our understanding of how regeneration is controlled under disease-mimicking conditions. For instance, Hyde and colleagues have shown that the extent of rod cell loss determines MG responsiveness: low rod cell loss inducing proliferation solely in "rod-committed" ONL precursors while extensive rod loss elicited responses in both ONL precursors and MG/MGPCs in the INL [54]. Moreover, cell-specific retinal cell ablation models have revealed that regenerative processes can exhibit "fate-bias", where newly generated cell fates favor the cell type that was lost [13–16]. Subtler fate biases follow even widespread retinal damage [63], suggesting that the regenerative process is informed by the nature of the injury/cell loss paradigm to guide retinal progenitor cell fate decisions. The sequence of temporally-delineated cell fates attending development appear to be recapitulated following widespread retinal injury [12]. However, how biases in fate decisions are controlled following either widespread or selective retinal cell loss, is unknown.

In mice, overexpression of neurogenic factors such as *ascl1* [4,5,7] in MG, and knockout of inhibitory factors such as the *nfi* genes [29] have been shown to promote limited regenerative responses. However, the neuronal cell types generated with these methods have been predominantly limited to bipolar-like cells [5], neither well matched to the cell types lost nor particularly relevant to retinal disease. However, inroads into producing more disease-relevant cell types such as retinal ganglion cells have recently been made by adding additional neurogenic factors such as *pou4f2* and *isl1* [7,64,65]. Increased understanding of mechanisms regulating fate-biased regeneration may help to promote the full regenerative potential of the mammalian retina. Here, we investigated transcriptomic responses to the selective loss of two "late born" retinal cell types [12], rod photoreceptors and bipolar interneurons, in an effort to reveal mechanisms that inform and regulate fate-biased regenerative mechanisms in the zebrafish retina.

We first characterized NTR-rod and NTR-bipolar transgenic lines following Mtz-induced cell loss, confirming the specificity of cell loss and quantifying proliferative responses in the INL and ONL (Fig 1). Our data showed proliferative responses in both the INL and ONL following rod ablation (Fig 1l), consistent with extensive rod cell loss activating MG, while bipolar ablation led primarily to INL proliferation (Fig 1m). Importantly, both paradigms induced INL proliferation at equivalent levels, suggesting comparable MG responses. Accordingly, the differences in gene expression observed between these paradigms cannot be explained solely by activation of ONL rod-committed precursors in the NTR-rod paradigm. We do, however, acknowledge that the differential involvement of rod precursors must account for some of the differences in gene expression we observe between paradigms. Additional variables that could lead to observed differences include: (1) relative expression levels, intracellular localization, and/or transgene copy number of NTR, the total number of cells lost, the genetic background of each line, and differential timing of cell loss and/or regeneration kinetics relative to sample collection timepoints. A particularly intriguing possibility is that different subpopulations of MG may exist that are dedicated to the regeneration of specific retinal cell types. Jusuf and

colleagues recently presented data consistent with this concept using single cell transcriptomics to suggest as many as six different MG subtypes exist in the zebrafish retina [66].

To comprehensively characterize transcriptomic changes during rod and bipolar cell regeneration, samples were collected across 12 time points encompassing cell loss and regeneration (Fig 2a). Prior microarray studies have successfully identified genes associated with MG activation [67] and networks activated following various retinal injury paradigms [62,68–72]. More recently, single-cell RNAseq (scRNA-seq) and ATAC-seq (Assay for Transposase-Accessible Chromatin) were used to compare light and NMDA induced retinal cell ablation paradigms during early phases of the regenerative process [29]. Here, by expanding the number of time points analyzed we sought to identify genes/pathways associated with: 1) inflammation and immune cell activation, 2) stem cell activation, 3) MG/MGPC proliferation, and 4) MGPC differentiation. We emphasized early time points (every 8 hr from 0–48 h) to identify genes linked to the initiation of fate-biased regenerative processes. A total of 364 shared, 918 rod-specific, and 1249 bipolar-specific DEGs were identified (Fig 2b). Overall, shared DEGs/pathways were limited more to early time points, while paradigm-specific DEGs/pathways were identified throughout the study. It is possible that there are more unaccounted for shared and unique DEGs that could arise from potential differences in the genetic makeup of the models. The number of DEGs diminished for both cell-types starting at 72 hpa. That paradigm-specific DEGs predominated supports the concept that injury/cell loss specifics inform the regenerative process [3,11–16].

KEGG pathway analysis per each timepoint identified previously implicated pathways (e.g., mTor, Notch, and Wnt signaling) [8,55,69,73,74], and terms linked to the regulation of stem cell function but not previously associated with retinal regeneration (e.g., glutathione metabolism [75], FoxO signaling [76], and cellular senescence [77]). To enhance pathway analyses, we grouped DEGs into four coordinately regulated sets for each paradigm: up or down early (16–32 hr) and up or down late (48–72 hr, Fig 3a and 3b). This accounted for the majority of DEGs and reinforced that paradigm-specific DEGs outnumbered shared DEGs (Fig 3c). Representative genes in the NTR-rod early up group included known factors (e.g., *activating transcription factor 3- atf3*, *stat3*) [62,78] and genes not previously associated with retinal regeneration (e.g., *glutathione S-transferase pi 1- gstp1*). Among the known set, factors associated with immune system reactivity were prevalent, including a number of genes previously shown to be upregulated in MG at 16h post light ablation in adult zebrafish [62] (e.g., *janus kinase 1- jak1*, *stat3*, *suppression of cytokine signaling 3b- socs3b*, *cytokine inducible SH2-containing protein- cish*, *irf9* and *complement component 7b- c7b*). GO analysis additionally identified shared late pattern enrichment for the broad term "regeneration" based on the differential regulation of the genes *annexin A1a- anxa1a*, *ubiquitin-like with PHD and ring finger domains 1- uhrf1*, *socs3a*, *capthespin Ba- ctsba*, *legumain- lgmn*, *major vault protein- mvp*, *v-fos FBJ murine osteosarcoma viral oncogene homolog Ab- fosab*, and *GLIS family zinc finger 3- glis3* (Fig 5). Next, IPA was used to reveal differential regulation of pathways at early and late stages of regeneration in each model. For both cell types, and at both early and late timepoints, multiple shared and unique immune-related pathways were identified, e.g., cytokines, JAK/STAT, Toll-like receptor and prolactin signaling (S15 Fig). We then produced unsupervised DEG networks using the coordinated expression pattern data, i.e., rod-early, rod-late, bipolar-early and bipolar-late. Known pro-regenerative and pro-inflammatory factors STAT3 and IL-6 [62,79] were identified as central mediators in the rod paradigm (Figs 4h and 5h), while PPARG and IRF7 were implicated as central mediators for the bipolar paradigm (Figs 4i and 5i). PPARG has recently been tied to zebrafish retinal progenitor cell function in adults through interactions with metabolic pathways [80]. Interestingly, IRF7 has been implicated as a key regulator of microglia reactivity, its upregulation promoting a phenotypic switch from an inflammatory to anti-

inflammatory state, aka stimulating resolution of microglia activation [81,82]. Our prior work suggested that accelerating the resolution process by exposure to the immunosuppressant dexamethasone 24 hrs after induction of rod cell ablation led to enhanced rod cell regeneration in zebrafish [83]. Together, these studies implicate induction of IRF7 expression as potential strategy for promoting neuronal regeneration.

Both IL-6 and interferons activate JAK/STAT signaling [79,84], a known mediator of the response to retinal damage [62] regulating cell migration [85], proliferation [62], apoptosis [86], immune system [87] and oxidative stress [88] responses. DEGs in the JAK/STAT family were primarily upregulated in the rod (e.g, *stat3*, *jak1*, *cish*, *socs1a/3a/3b*) and bipolar paradigms (e.g, *cish*, *socs1a/3a/3b*, *stat1a/1b/2/5a/5b*). However, the specific JAK/STAT family members involved differed between our two paradigms (e.g., *stat3* for rod and *stat1a/b*, *stat2 and stat5a/b* for bipolar). Similarly, different cell death pathways were also implicated, NTR-rod being linked to necroptosis (Fig 4b) while NTR-bipolar was linked to apoptosis (Fig 4c and 4f), These findings align with a recent drug screen in the same NTR-rod model where inhibitors and genetic disruption of parthanatos and necroptosis pathways protected rods from cell death while inhibition of apoptosis did not [41]. Interestingly, recent work in multiple mouse tumor models links necroptotic signaling to the activation of IL-6 and STAT3 [89–91] while apoptotic signaling was linked to STAT5 activation [92], matching gene expression and cell death patterns in our models. JAK/STAT signaling receives feedback inhibition from Suppression of Cytokine Signaling (SOCS) genes [93]. Notably, ablation of both cell types was followed by upregulation of *socs3a*, *socs3b*, *socs1a* and *cish*. *Socs3a* has been implicated as a mediator of hair and liver cell regeneration in zebrafish through activation of resident stem cells and progenitor differentiation [94,95], and is strongly upregulated in an adult zebrafish photoreceptor lesion model [70].

CRISPR/Cas9 mediated disruption of *stat3* led to significant reductions in proliferation and rod photoreceptor regeneration kinetics (Fig 6), consistent with a reduction in photoreceptor regeneration following light-damage in *stat3* morpholino injected eyes [62]. In contrast, disruption of *txn* (*thioredoxin*), *stat5a/b* and *c7b* (*complement factor 7b*) led to enhanced rod regeneration. *C7b*, *txn* and *stat5a/b* are each key immune system factors, and chromatin data from the ENCODE project identified *c7b* and *txn* as target genes of the *Stat5* transcription factor in humans [96]. Mutation of these factors can now be added to a growing list of immuno-modulatory techniques that enhance rod photoreceptor regeneration in the larval zebrafish retina [39,58,97]. When combined with our observations of robust differences in immune system activation during regeneration of rods or bipolar cells, we interpret this to indicate that the immune system is a key regulator of fate-biased regeneration. Further insights into the regulation of fate-biased regenerative mechanisms will likely aid in the development of cell-type targeted therapeutic strategies to restore vision.

## Materials and methods

### Ethics statement

All procedures were completed in compliance with animal care and use protocols approved by both the Medical College of Georgia (now Augusta University) Institutional Animal Care and Use Committee (IACUC), as well as the Johns Hopkins University Animal Care and Use Committee (ACUC).

### Aquaculture

Zebrafish were maintained under standard environmental conditions (~28°C, 14hr:10hr light:dark cycle). Larvae were fed paramecia starting at 5 days post-fertilization (dpf), then

paramecia and *Artemia salina* from 9–16 dpf. Late stage larvae (e.g., 8 dpf) slated for imaging were given 1-phenyl 2-thiourea (PTU) beginning at 1 dpf and kept in 12 well plates in 5 ml of system water until 12 dpf after which they were sacrificed.

## Transgenic lines

A bipartite Gal4/UAS [98] transgenic zebrafish line derived with a 1,482 bp promoter element from the *nyctalopin* (*nyx*) gene [99,100], *Tg(nyx:Gal4-VP16)q16a*, and a Gal4-driven reporter, *(UAS:GAP-YFP)q16b*, expresses membrane-tagged yellow fluorescent protein (YFP) in a sub-population of bipolar cells [48]. Crosses to a UAS-reporter/effector line facilitating prodrug-induced cell ablation, *Tg(UAS-E1b:NfsB-mCherry)c264* [101], resulted in triple transgenic fish expressing an NTR-mCherry fusion protein in subsets of *nyx*-promoter targeted bipolar cells (*q16a;q16b;c264*; hereafter, NTR-bipolar lines). A transgenic line derived with a 3.7 kb promoter element from the *rhodopsin (rho)* gene (kind gift of Dr. Shoji Kawamura) [49] expresses a YFP-NTR fusion protein in rod photoreceptor cells, *Tg(rho:YFP-Eco.NfsB)gmc500* [46] (hereafter, NTR-rod lines). Retinal morphology was monitored during imaging studies by crossing both NTR lines with *Tg(pax6-DF4:gap43-CFP)q01* which expresses membrane tagged CFP throughout the retina (see Fig 1A and 1B) [102]. Transgenic fish were propagated in the *roy orbison* (aka, *mpv17$^{a9/a9}$*, hereafter *roy$^{a9/a9}$*) [103,104] pigmentation mutant background to facilitate intravital imaging [105]. To minimize potential effects of genetic background regular outcrosses to an AB wildtype strain (WT) and *roy* lines from other sources were performed to increase genetic diversity.

## Mtz treatment

For histological experiments, *in vivo* imaging experiments and all microarray tissue collections, transgenic NTR-bipolar and NTR-rod larvae were screened and separated into two equal sized groups at 6 dpf: 1) non-treated controls (Cntl); and 2) those treated with 10mM Mtz for 24 hr to induce cell death, from 6–7 dpf (or DMSO-treated controls where applicable). After Mtz treatment, fish were rinsed into system water and fed paramecia until sacrificed.

## mRNA isolation and preparation

Whole eye mRNA was prepared across twelve time points: t0 (just prior to Mtz exposure); t8, t16, and t24 hr (during Mtz treatment); and t32, t40, t48, t60, t72, t96, t144, and t240 hr (after Mtz removal during recovery (Fig 2A). Sample collection started at 8:00 a.m. on the day larvae reached 6 dpf. At each time point, 16 eyes from 8 larvae were isolated per condition. To isolate eyes, larvae were euthanized in 0.4% tricaine for 5 mins followed by cervical transection. Whole eye pairs were manually isolated under an Olympus SZX16 epifluorescence stereoscope and immediately placed into a chilled/sterile 1.5 ml tube with 50 µl of TriZOL (Life Technologies); samples were then stored at -80˚C until extraction. This protocol was followed for three biological repeats per condition/time point for NTR-rod and NTR-bipolar lines. Samples were batch extracted and processed following the final time. All samples were processed in the Medical College of Georgia (MCG, former Mumm lab location) Integrated Genomics Core for mRNA isolation, cDNA library construction, and microarray hybridization.

## Microarrays

GeneChip Zebrafish Genome Microarrays (Affymetrix) were processed by the MCG Genomic Core. Each run involved sibling zebrafish and consisted of 23 RNA samples across each regenerative paradigm—1 pretreatment, and 11 time points from the control and Mtz treatment

groups. RNA sample quality was assessed by concentration and the RNA Integrity Number (RIN), an algorithm that assigns integrity values to RNA measurements [106]. Only samples attaining a minimal concentration of $\geq$100ng/$\mu$L (5$\mu$L total) and a RNA integrity number (RIN) value of $\geq$8 were used for microarray hybridizations [106]. Runs where a minimum of 21 of 23 samples were deemed of good quality were utilized in the microarray studies. Three runs were processed per regenerative paradigm resulting in six microarray sets (S1 Table) and a total 138 microarrays overall.

## Tissue preparation and immunohistochemistry

For immunohistochemistry, larval zebrafish were euthanized, fixed in 4% paraformaldehyde (PFA) for 8 hr, washed three times in 1x PBS (phosphate buffered saline; EMD Millipore) for 30 min and stored at 4˚C. Samples were mounted in cryogel embedding medium within the next 2–4 days, frozen in liquid nitrogen then stored at -80˚C until sectioned in the lateral plane at 25 μm thickness with a cryostat. Sliced sections were collected on standard microscope slides and stored at 4˚C.

For immunolabeling, slides were air dried at room temperature for ~1 hr, rinsed in 1xPBS and then re-fixed with 4% PFA for 15 min. PBST rinses (1xPBS +0.1% Tween20, Fisher Scientific) were conducted to remove trace PFA followed by 5 min antigen retrieval with SDS (1% Sodium Dodecyl Sulfate; Fisher Scientific) in PBS. The blocking phase was performed with 3% goat serum in PBDT (1xPBS, 1% BSA, 1% DMSO, 0.1% TritonX-100) for 30 min and incubated with primary antibody/1% goat serum/PBDT overnight at 4˚C. The next morning, slides were rinsed in PBST, stained with secondary antibody/PBDT ~1 hr in a light protected humidity chamber and cover-slipped (22x50 mm, Fisher Scientific). PBST rinses removed unbound secondary antibody. Samples were protected with Vectashield + DAPI (Vector Laboratories) and cover-slipped (24x50 mm, Fisher Scientific).

Primary antibodies included: mouse anti-PCNA monoclonal antibody (1:1000, clone PC10; Sigma Aldrich), rabbit anti-Caspase-3 monoclonal antibody (1:500, clone C92-605; BD Biosciences), Click-iT Tunel Alexa Fluor 647 (1:500; Life Technologies), mouse anti-zpr-1 monoclonal antibody (1:750; ZIRC), mouse anti-zpr-3 monoclonal antibody (1:750; ZIRC), mouse anti-ZS-4 monoclonal antibody (1:750; ZIRC). Secondary antibodies included: anti-mouse Alexa Fluor 430 (1:500; Life Technologies), anti-mouse Alexa Fluor 635 (1:500; Life Technologies), anti-rabbit Alexa Fluor 430 (1:500; Life Technologies).

## Cell counting and analysis

Images were collected with an Olympus FV1000 Confocal Microscope (405, 440, 488, 515, 559, and 635nm laser lines). Stacked confocal images were obtained using a 40x oil immersion objective with a 2.5 μm step size, 130 μm aperture, and 10 μm total depth. Five sections were collected per retina centered around the optic nerve. Images (Olympus.OIB format) were analyzed using ImageJ. Manual cell counts were averaged across sections per retina and averaged within each group. For statistical comparisons, Student's t test was used to assess paired conditions while Welch's one-way ANOVA followed by Dunnett's test for multiple comparisons was used to assess differences across more than two conditions. Results from all statistical comparisons are shown graphically.

## Sample quality and differential expression analysis

Data were collected from 138 Genechip Zebrafish Genome arrays (Affymetrix) across four conditions: NTR-rod cntrl, NTR-rod Mtz, NTR-bipolar cntrl, and NTR-bipolar Mtz. The Bioconductor suite in the R statistical software [107], LIMMA [108], Carmaweb [109], and

Microsoft Access were used for data processing, differential expression testing and storage. Prior to statistical analysis, Robust Multi-array Average (RMA) pre-processing routine [110] and Quality control assessments [111] were used for background correction, normalization and quality control. A set of differentially expressed genes was derived using LIMMA with a false discovery rate adjusted p-value of 0.05. Differentially expressed genes were defined as those expressing a 1.5-fold up or downregulation at any time point in regeneration. Pathway analysis was performed using QIAGEN Ingenuity Pathway Analysis (IPA), Gene Ontology (GO) and KEGG pathway analysis (via the pathfindR R studio package). IPA was also utilized to generate gene networks. An adjusted p value below 0.05 was used throughout as a cutoff for significance in all analyses.

## qPCR processing

Extracted mRNA samples were reverse transcribed (Qiagen Omniscript RT kit, Qiagen) and stored at -20˚C. Samples were run in triplicate using the BioRad iQ SYBR Green Supermix (BioRad) in iCycler IQ 96 well PCR plates (Bio-Rad) on a BioRad iCycler equipped with an iCycler iQ Detection System. The protocol consisted of three phases: 1) 92˚C for 10:00 min, 2) 50x 92˚C for 00:15 min, 60˚C for 01:00 min, 3) 81x 55˚c➡95˚c for 00:10 min. β-actin served as the house keeping gene and $2^{-\Delta\Delta CT}$ method was used for normalization to ensure equal amounts of cDNA for comparisons. qPCR primers were designed using the online tool QuantPrime.

## CRISPR/Cas9 mediated targeting

CRISPR/Cas9 mediated redundant targeting injections were performed at the one-cell stage of gmc500 embryos utilizing the published strategy and gRNA table by Wu et al [59]. Published sgRNAs for *stat3*, *c7b*, *txn*, *prdm1a*, *pparg*, *stat5a/b*, and *nfia* were ordered as DNA oligos, assembled with the general CRISPR tracr oligo, and then transcribed using pooled in vitro transcription (HiScribe T7 High Yield RNA Synthesis kit, New England BioLabs) and cleaned up with the NEB Monarch RNA Cleanup kit. A mixture of all four sgRNAs (1 ng in total) and Cas9 protein (2.5 μM, IDT) was injected into *rho*:*YFP-NTR* embryos at the one-cell stage for targeting each gene.

## Quantitative real-time polymerase Chain reaction (qRT-PCR)

To validate *stat3* mutation by CRISPR/Cas9 redundant targeting, 3 groups of 10 larvae of both uninjected and 4xgRNA injected transgenic larvae at 2 dpf were processed through a previously published protocol to measure gene expression with qRT-PCR [41]. Briefly, RNA was purified using NEB Monarch RNA Cleanup kit and reverse transcribed to cDNA using qScript cDNA synthesis kit (QuantaBio). Quantitative PCR was conducted using designed primers with the primerdb database and PowerUp SYBR Green Master Mix (ABI) in QuantaStudio (ABI). ΔΔCT analysis was performed to calculate relative fold change in gene expression levels between Cas9/4xgRNA injected larvae and controls. Each experiment was performed in triplicate.

## ARQiv scans to measure rod photoreceptor development, loss, and regeneration

Wildtype non-injected and target sgRNA injected *Tg(rho*:*YFP Eco. NfsB)gmc500* larvae were treated with Mtz at 5 dpf and subsequently rod cell loss or regeneration was measured at 7 dpf and 9 dpf, respectively, using the ARQiv system, as previously described [39]. Effects on rod

cell development were measured at 5 dpf in mutant groups that showed significantly enhanced rod regeneration.

## Supporting information

**S1 Fig. Confirmation of NTR-rod transgene specificity.** (a-a") Histological staining in the NTR-rod model showing transgenic Rho:YFP-NTR expression (green) along with antibody staining for rod outer segment marker Zpr-3 (purple). (b-b") Histological staining in the NTR-rod model showing transgenic Rho:YFP-NTR expression (green) along with antibody staining for cone marker Zpr-1 (purple).
(TIF)

**S2 Fig. Additional in vivo imaging and TUNEL/PCNA histological imaging.** (a-b series) In vivo time-series imaging following NTR-rod larvae that were uninjured (a-a") or received Mtz (b-b"). Images were taken in the same fish at 6 dpf (before Mtz onset in +Mtz larvae), 7 dpf (after Mtz removal) and 11 dpf (following recovery). (c-d series) In vivo time-series imaging following NTR-bipolar larvae that were uninjured (c-c") or received Mtz (d-d"). Images were taken in the same fish at 6 dpf (before Mtz onset in +Mtz larvae), 7 dpf (after Mtz removal) and 11 dpf (following recovery). Larvae for each ablation paradigm express CFP derived from Tg(pax6-DF4:gap43-CFP)q01 to label general retinal structures (a-d series). (e-f series) Histological staining in uninjured and +Mtz eyes at 7 dpf for TUNEL in the NTR-bipolar (e-e') and NTR-rod (f-f') paradigms. (g-h series) Histological staining in uninjured and +Mtz eyes at 9 dpf for PCNA in the NTR-bipolar (g-g') and NTR-rod (h-h') paradigms.
(TIF)

**S3 Fig. qRT-PCR validation of mRNA expression changes.** (a, b) A subset of tested genes were selected for validation using quantitative Reverse Transcription–Polymerase Chain Reaction (qRT-PCR). 8 genes were selected for a temporal analysis from 0-72h post Mtz in either the NTR-rod (A) or NTR-bipolar (B) paradigm with beta actin serving as a control.
(TIF)

**S4 Fig. Analysis of DEGs and associated pathway terms at timepoint 8 hours following Mtz.** (a) Volcano plot showing the identified up (orange) and down (blue) DEGs in both NTR-rod and NTR-bipolar treatment paradigms. (b) Venn diagram showing the unique as well as shared (and % of shared) DEGs between the two paradigms. (c) KEGG pathfindR identified significantly enriched pathways in the NTR-rod and NTR-bipolar paradigms.
(TIF)

**S5 Fig. Analysis of DEGs and associated pathway terms at timepoint 16 hours following Mtz.** (a) Volcano plot showing the identified up (orange) and down (blue) DEGs in both NTR-rod and NTR-bipolar treatment paradigms. (b) Venn diagram showing the unique as well as shared (and % of shared) DEGs between the two paradigms. (c) KEGG pathfindR identified significantly enriched pathways in the NTR-rod and NTR-bipolar paradigms as well as those shared between the two paradigms.
(TIF)

**S6 Fig. Analysis of DEGs and associated pathway terms at timepoint 24 hours following Mtz.** (a) Volcano plot showing the identified up (orange) and down (blue) DEGs in both NTR-rod and NTR-bipolar treatment paradigms. (b) Venn diagram showing the unique as well as shared (and % of shared) DEGs between the two paradigms. (c) KEGG pathfindR identified significantly enriched pathways in the NTR-rod and NTR-bipolar paradigms as well as

those shared between the two paradigms.
(TIF)

**S7 Fig. Analysis of DEGs and associated pathway terms at timepoint 32 hours following Mtz.** (a) Volcano plot showing the identified up (orange) and down (blue) DEGs in both NTR-rod and NTR-bipolar treatment paradigms. (b) Venn diagram showing the unique as well as shared (and % of shared) DEGs between the two paradigms. (c) KEGG pathfindR identified significantly enriched pathways in the NTR-rod and NTR-bipolar paradigms.
(TIF)

**S8 Fig. Analysis of DEGs and associated pathway terms at timepoint 40 hours following Mtz.** (a) Volcano plot showing the identified up (orange) and down (blue) DEGs in both NTR-rod and NTR-bipolar treatment paradigms. (b) Venn diagram showing the unique as well as shared (and % of shared) DEGs between the two paradigms. (c) KEGG pathfindR identified significantly enriched pathways in the NTR-rod and NTR-bipolar paradigms as well as those shared between the two paradigms.
(TIF)

**S9 Fig. Analysis of DEGs and associated pathway terms at timepoint 48 hours following.** (a) Volcano plot showing the identified up (orange) and down (blue) DEGs in both NTR-rod and NTR-bipolar treatment paradigms. (b) Venn diagram showing the unique as well as shared (and % of shared) DEGs between the two paradigms. (c) KEGG pathfindR identified significantly enriched pathways in the NTR-rod and NTR-bipolar paradigms.
(TIF)

**S10 Fig. Analysis of DEGs and associated pathway terms at timepoint 60 hours following Mtz.** (a) Volcano plot showing the identified up (orange) and down (blue) DEGs in both NTR-rod and NTR-bipolar treatment paradigms. (b) Venn diagram showing the unique as well as shared (and % of shared) DEGs between the two paradigms. (c) KEGG pathfindR identified significantly enriched pathways in the NTR-rod and NTR-bipolar paradigms as well as those shared between the two paradigms.
(TIF)

**S11 Fig. Analysis of DEGs and associated pathway terms at timepoint 72 hours following Mtz.** (a) Volcano plot showing the identified up (orange) and down (blue) DEGs in both NTR-rod and NTR-bipolar treatment paradigms. (b) Venn diagram showing the unique as well as shared (and % of shared) DEGs between the two paradigms. (c) KEGG pathfindR identified significantly enriched pathways in the NTR-bipolar paradigm.
(TIF)

**S12 Fig. Analysis of DEGs and associated pathway terms at timepoint 96 hours following Mtz.** (a) Volcano plot showing the identified up (orange) and down (blue) DEGs in both NTR-rod and NTR-bipolar treatment paradigms. (b) Venn diagram showing the unique as well as shared (and % of shared) DEGs between the two paradigms.
(TIF)

**S13 Fig. Analysis of DEGs and associated pathway terms at timepoint 144 hours following Mtz.** (a) Volcano plot showing the identified up (orange) and down (blue) DEGs in both NTR-rod and NTR-bipolar treatment paradigms. (b) Venn diagram showing the unique as well as shared (and % of shared) DEGs between the two paradigms. (c) KEGG pathfindR identified significantly enriched pathways in the NTR-rod paradigm.
(TIF)

**S14 Fig. Analysis of DEGs and associated pathway terms at timepoint 240 hours following Mtz.** (a) Volcano plot showing the identified up (orange) and down (blue) DEGs in both NTR-rod and NTR-bipolar treatment paradigms. (b) Venn diagram showing the unique as well as shared (and % of shared) DEGs between the two paradigms. (c) KEGG pathfindR identified significantly enriched pathways in the NTR-rod and NTR-bipolar paradigms.
(TIF)

**S15 Fig. Immune related terms identified from IPA for NTR-rod early pattern.** (a-d) IPA was used to test for enriched immune related pathway terms from DEGs for the NTR-rod early pattern.
(TIF)

**S16 Fig. Immune related terms identified from IPA for NTR-bipolar early pattern.** (a-d) IPA was used to test for enriched immune related pathway terms from DEGs for the NTR-bipolar early pattern.
(TIF)

**S17 Fig. Immune related terms identified from IPA for NTR-rod late pattern.** (a-d) IPA was used to test for enriched immune related pathway terms from DEGs for the NTR-rod late pattern.
(TIF)

**S18 Fig. Immune related terms identified from IPA for NTR-bipolar late pattern.** (a-d) IPA was used to test for enriched immune related pathway terms from DEGs for the NTR-bipolar late pattern.
(TIF)

**S19 Fig. Literature search for immune or regeneration related terms.** (a-b) The top ~50–60 DEGs for the NTR-rod (a) and NTR-bipolar (b) patterns were searched in the literature for any links to a role in the immune system or in regeneration. The percentage of those with a link to each term are shown. (c-d) Bar showing the percentage of top DEGs that were implicated in the immune system (c) or regeneration (d).
(TIF)

**S20 Fig. Effects on rod cell development and ablation in *txn*, *stat5a/b* and *c7b* crispants.** (a) Quantification of rod cell development pre-ablation by plate reader assay at 5 dpf in wildtype (WT) and MUT larvae for target genes *txn*, *stat5a/b* and *c7b*. (b) Quantification of rod cell ablation following Mtz treatment by plate reader assay at 7 dpf in non-ablated (-Mtz) and ablated controls (+Mtz) and ablated MUT larvae for target genes *txn*, *stat5a/b* and *c7b*. For statistical comparisons, Welch's one-way ANOVA was followed by student's t test with Dunnett's method for multiple comparisons correction. Asterisks indicate the following p-value ranges: * = $p < 0.05$, ** = $p < 0.01$, *** = $p < 0.001$, and **** = $p < 0.0001$, "ns" indicates $p > 0.05$.
(TIF)

**S1 Table. mRNA quality of samples collected for microarray analysis.**
(XLSX)

**S2 Table. qPCR primers used.**
(XLSX)

**S3 Table. Early pattern genes for bipolar and rod ablation paradigms.**
(XLSX)

**S4 Table. Late pattern genes for bipolar and rod ablation paradigms.**
(XLSX)

**S5 Table. Gene Ontology (GO) pathway analysis results for early and late patterns.**
(XLSX)

**S6 Table. Ingenuity Pathway Analysis (IPA) results for early and late patterns.**
(XLS)

**S7 Table. All differentially expressed genes identified in NTR:rod paradigm.**
(XLSX)

**S8 Table. All differentially expressed genes identified in NTR:bipolar paradigm.**
(XLSX)

## Acknowledgments

We thank BioRender.com for the permission to use scientific icons in making figures.

## Author Contributions

**Conceptualization:** Steven L. Walker, Jeff S. Mumm.

**Data curation:** Kevin Emmerich, Steven L. Walker, David T. White, Anneliese Ceisel, Yong Teng.

**Formal analysis:** Kevin Emmerich, Steven L. Walker, Guohua Wang, Fang Wang, Yong Teng, Zeeshaan Chunawala, Gianna Graziano, Saumya Nimmagadda, Meera T. Saxena, Jiang Qian, Jeff S. Mumm.

**Funding acquisition:** Jeff S. Mumm.

**Investigation:** Kevin Emmerich, Steven L. Walker, Guohua Wang, David T. White, Anneliese Ceisel, Fang Wang, Yong Teng, Zeeshaan Chunawala, Gianna Graziano, Saumya Nimmagadda, Meera T. Saxena, Jiang Qian, Jeff S. Mumm.

**Methodology:** Steven L. Walker, David T. White, Yong Teng, Jeff S. Mumm.

**Project administration:** Jeff S. Mumm.

**Resources:** Jeff S. Mumm.

**Supervision:** Jiang Qian, Jeff S. Mumm.

**Validation:** Kevin Emmerich, Steven L. Walker, David T. White, Anneliese Ceisel, Yong Teng.

**Visualization:** Kevin Emmerich, Steven L. Walker, David T. White, Jeff S. Mumm.

**Writing – original draft:** Kevin Emmerich, Steven L. Walker, Jeff S. Mumm.

**Writing – review & editing:** Kevin Emmerich, Anneliese Ceisel, Zeeshaan Chunawala, Saumya Nimmagadda, Meera T. Saxena, Jeff S. Mumm.

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
