## [Decision Letter · Decision Letter 0]

21 Mar 2023

Dear Dr Mumm,

Thank you very much for submitting your Research Article entitled 'Transcriptomic comparison of two selective retinal cell ablation paradigms in zebrafish reveals shared and cell-specific regenerative responses' to PLOS Genetics.

The manuscript was fully evaluated at the editorial level and by independent peer reviewers. The reviewers appreciated the attention to an important problem, but raised some substantial concerns about the current manuscript. Based on the reviews, we will not be able to accept this version of the manuscript, but we would be willing to review a much-revised version. We cannot, of course, promise publication at that time.

If you decide to revise the manuscript for further consideration at PLOS Genetics, please aim to resubmit within the next 60 days, unless it will take extra time to address the concerns of the reviewers, in which case we would appreciate an expected resubmission date by email to plosgenetics@plos.org.

We are sorry that we cannot be more positive about your manuscript at this stage. Please do not hesitate to contact us if you have any concerns or questions.

Yours sincerely,

Gregory Barsh

Editor-in-Chief

PLOS Genetics

Gregory Copenhaver

Editor-in-Chief

PLOS Genetics

Reviewer's Responses to Questions

**Comments to the Authors:**

Reviewer #1: In the article by Emmerich, et al., the Authors leverage prior technologies created in the laboratory of Dr. Mumm to investigate transcriptomic changes across the regeneration of the zebrafish retina in two different, cell type-specific ablation paradigms. This work assesses the functional consequence of loss of individual cell fates, serving as a proxy to degenerations that affect the human retina and providing insights into the gene regulatory networks that promote cell-type specific regeneration in zebrafish and/or limit the regenerative ability of the human retina.

The manuscript does a good job of providing details of background and rationale for the studies, technologies utilized, interpretations of transcriptomic and pathway analyses, and providing context of the discoveries within prior research.

Additionally, the Authors provide biological relevance for findings of stat3 differential expression.

Areas to be addressed:

Minor Points:

1) Fig. 1 - l and m - please label the figure with the treatment paradigm that each panel represents

2) Statistics:

- All figures with statistical analyses: please provide the statistical test utilized for each analysis within the figure legend where statistical data is presented. It is unclear which panels utilized t-tests versus One-way ANOVAs based on the presented data/methods

- no statistics on figure 6 c

- line 230-233 - the authors make comparisons between differences in location of proliferative cells between rod and bipolar ablation paradigms. However, no such statistical comparison is actually presented.

- Appropriateness of the statistical methods utilized. Please provide rationale for the use One-Way ANOVA (Figure 1l-m; Figure 6 i). In these analyses, the effect of 2 different variables is being tested; either Region (rod precursor or Muller glia proliferation) and MTZ treatment (Figure 1); or Region (rod precursor or Muller glia proliferation) and stat3 expression (Figure 6). It seems like a two-way ANOVA would be more appropriate.

3) KEGG pathway analyses would be more informative if data was presented with up- and down-regulated genes separated. For example, in Supp. Fig 4C - KEGG pathway analysis of ALL differential genes shows an 'enrichment' of phototransduction genes. It would be more informative if all down-regulated genes were analyzed indepedently of 'up-regulated' genes and 'phototransduction' was significantly enriched, consistent with loss of rod photoreceptors. Analyzing both up and down-regulated genes together may actually be reducing the significance of pathways that are affected.

4) Data availability - Unclear if the Microarray data has been submitted to a publicly available resources (GEO)

Major Point:

The goal of the study is to provide evidence of differential regenerative responses at the transcriptomic level based on specific cell type ablation paradigms. This goal is met through the comprehensive microarray analyses. However the biological relevance of these findings remains limited.

stat3 has already been implicated in regulating regenerative responses in both fish and mice, a finding acknowledged by the Authors with appropriate references. To achieve a goal of identifying cell type-specific ablation responses, the authors would provide one or both of the following:

1) Proliferative response in bipolar ablation paradigm with/without loss of stat3; direct comparisons to rod ablation response. If stat3 is most important for Muller glia-derived regeneration, one may anticipate a more drastic inhibition of bipolar cell regeneration. An observation of no-difference compared to rod-ablation would suggest that stat3 is required for both rod precursor and Muller glia-derived regeneration, supporting the transcriptomic data where stat3-signaling was affected in both paradigms

2) Examination of the regenerative responses when a bipolar-cell ablation 'hub' gene is reduced (PPARG), in both the rod and bipolar ablation paradigms. The differential activity of the KEGG pathway analyses might suggest differential response.

It should be noted that any result from either/both of these suggested experiments would be informative.

Reviewer #2: Kevin Emmerich and colleagues use microarray transriptomics to characterize homogenates of eye tissue in a time-course study where wither rods were ablated (rod-NTR) or a small subset of bipolar cells were ablated (bipolar-NTR). The responses of the transcriptome were different. Some surprises in the data were that these differences began immediately, which the authors interpret as differences in fate-choice. Some softening of that interpretation (offering alternatives) is suggested below.

Considering the potential for stem-cell-based regenerative therapies in blindness (which is the most likely pioneer for such treatments of other neurodegenerative diseases), the massive data sets produced here are an excellent resource for the community. Their comparative framing (between regeneration paradigms, across time) adds considerably to their anticipated power. It is reasonable to expect that mining these data sets will influence future researchers as they seek to implement plans to therapeutically regenerate cells.

If the work to be repeated today it would consider single cell transcriptomics and allow an additional depth of information, but the data provided here are valuable nonetheless. Some ideas about the manuscript appear below, focussed on alternative interpretations the authors might wish to consider.

1) Line 148 & opening of Discussion & closing of Discussion (line 707), regarding the suggestion that the type of cell loss informs the MG stem cell activation via altering *cell-fate* choices. The writing seems to have an unstated assumption that only one population of MG stem cells is present, and only one population is activated regardless of ablation paradigm, and thus the differential outcomes thus must result from differential fate choices. It is important to forthrightly consider an alternative interpretation that multiple populations of stem cells exist, and actually your paradigms are triggering different ones, thereby producing distinct gene abundances at initiation. The wording in abstract (line 49, 57) seems more appropriate, i.e. agnostic to ‘how’ the response is different, and instead identifying that it is different.

a. Data in Fig 1 (described in lines 227-238) undercut the interpretation that differences in the response are based in cell fate choices, an interpretation that would require the same stem cell population to be activated towards two different states. Instead, what is happening here is that two different stem cell populations are activated. [still very interesting, but must be interpreted with care; and the Rationale that the data give you instructions to replace a particular cell type then seems confounded too]

2) Discussion should be forthright in describing alternative reasons why one might observe differential responses other than just a difference between cell types being ablated.

a. E.g. a technical artefact could arise if the two lines have different NTR amount or different NTR intracellular-localization, or presence/absence of Gal4-VP16 – any of these could alter the quality of cell death and signals produced even within a given cell type.

b. The impact of ablating ~hundred cells vs. thousands of cells (in bipolar vs. rod lines) would be expected to be substantial.

c. Moreover, the random insertion of multiple transgenes differs in the two lines (and likely includes many haplotype differences dragged along with the transgenes in the two lines) such that the genetic background of the compared fish is probably quite different [consider that the standard in mice is to now compare multiple isogenic lines for any transgenic-based phenotype].

d. It is also possible that the peak of cell death might be offset by several hours in one line vs. the other, due to the Tg or the inherent resilience of the cells, and so the responses would occur in distinct time bins within the circadian cycle.

3) The beginning of the Results has a focus on specificity of the induced cell death (e.g. line 155). The compelling data to challenge this claim must exist within the transcriptomic data sets. E.g. rod opsin and transduction machinery ought to be reduced in abundance for the NTR-rod line compared to controls and compared to the NTR-bipolar line. Bipolar-specific genes should show the opposite pattern. A comment on this seems worth including in this early section of the Results (even if it is to offer the logic and send the reader to later data).

a. This is alluded to later (line 430) but not with enough detail to serve the purpose requested here. What are some examples of genes and what is their differential abundance? [image resolution in Fig 3b-h is poor, and I cannot read any gene names]

4) Approriate validation of this transciptomics approach, considering the presented rationale, would be to show therapeutic potential – i.e. that providing one of the DEG is sufficient to enhance the regeneration. Instead, the authors chose to demonstrate that a DEG, stat3, is required for regeneration. Thus the path to translation has not been clearly established here, though the data is undoubtedly of theoretical interest for that purpose.

Minor suggestions:

a) I’m baffled by how the in vivo imaging was accomplished (it is excellent). How does one image through a pigmented eye in a living 11 dpf zebrafish? (roy mutants are still pigmented). I could not find any of this in the Methods or associated Results near description of Figure 1.

b) where does the CFP in your images come from (Fig 1 legend should indicate this).

c) Line 35 ‘in’ s/b ‘and’

d) Line 70 you implicate immune-related genes, not the immune system itself. It’s only “the immune system” if you accept the over-simplistic assumptions of a KEGG pathway as top-line gene function assignments. “strongly implicated” seems too strong with the adjacent current wording. But I think the authors can be given licence to interpret their data as they prefer, where it is obvious that it is interpretation-not-statement-of-fact.

e) Related to above comment – line 514: is 38% and 44% an impressive number? I think most any DEG data set might show a similar type of abundance of genes related to the immune system. What if you compare the control fish from 6dpf and 11 dpf, is the abundance of these ‘immune-related genes’ <25%? Maybe not an analysis worth doing, but I think you could soften your language: you’ll always find these associations if you go looking for them.

f) Line 117, (& 149, 61, 588). “Retinal degenerative diseases typically involve the loss of discrete cell types.” A version of this statement was made earlier too – it seems a bit simple, self-serving, and certainly needs some references and examples. E.g. many blinding disorders involve loss of function mutations that may not (for many years) lead to cell loss (maybe they produce dysmorphic cells). Are there prominent examples of disorders where bipolar cells are suddenly lost? To be more optimistic in my comment, learning how to regenerate a rod photoreceptor is valuable as an imagined therapy for many many forms of blindness, including those where more-than-rods are dying. So it seems like the writing leans on this rationale too hard.

g) Line 184 delete ‘selective’. Nothing here demonstrates the ablation is selective (i.e. maybe other cells are killed too).

h) Line 197 to allow Figures to be interpretable as stand-alone items, define and/or explain “NTR”.

**Have all data underlying the figures and results presented in the manuscript been provided?**

Reviewer #1: Yes

Reviewer #2: **No: **Authors describe this in the future tense: "Raw transcriptomic data **will** be made available in a repository before publication"

PLOS authors have the option to publish the peer review history of their article (what does this mean?). If published, this will include your full peer review and any attached files.

Reviewer #1: No

Reviewer #2: No

---

## [Decision Letter · Decision Letter 1]

18 Jul 2023

Dear Dr Mumm,

Thank you very much for submitting your Research Article entitled 'Transcriptomic comparison of two selective retinal cell ablation paradigms in zebrafish reveals shared and cell-specific regenerative responses' to PLOS Genetics.

The revised manuscript was seen by the previous reviewers, both of whom are positive. There are some remaining concerns raised by reviewer #1 (below) that we ask you address in a hopefully final round of minor revision that will not necessarily require additional external review. 

We therefore ask you to modify the manuscript according to the review recommendations. Your revisions should address the specific points made by each reviewer.

Yours sincerely,

Gregory S. Barsh

Editor-in-Chief

PLOS Genetics

Gregory Copenhaver

Editor-in-Chief

PLOS Genetics

Reviewer's Responses to Questions

**Comments to the Authors:**

Reviewer #1: The authors have sufficiently addressed the previous critiques. However, there are two areas of the manuscript that need to be addressed:

1) “In contrast, bipolar loss led solely to increased INL proliferation” – statistics don’t support this conclusion; only indicate that total proliferation is increased. INL proliferation is listed as 'ns' compared to control

2) CRISPants (Figure 7). While the inclusion of this data was recognized as a response to both reviewers, the data do not fully support the Authors' conclusions as presented. The Authors state that "Of the six factors tested (Fig. 7b-c), mutation of three led to enhanced rod regeneration (from least to greatest effect: txn (+70%), stat5a/b +85%), and c7b (+96%), while three had no statistically significant effect (nfia, pparg, prdm1a)" compared to WT animals. While txn, stat5a/b and c7b MUT crispants do seem to have an increase in NTR:YFP compared to WT, other possible explanations need to be addressed. Are rod cell numbers equivalent at the start of the injury paradigm in each mutant condition? Are rod cells more resistant to MTZ-mediated cell death when these specific genes are mutated? Other more nuanced scenarios could also explain the observed results. The Authors should at least examine the effect of each mutation on general retinal development (as they are germline mutations) and acknowledge alternative mechanisms exist that lead to what the Authors interpret as 'enhanced regeneration'.

Reviewer #2: I think this is the best job of addressing Reviewer comments I have seen. Congrats to all the authors on a great contribution!

**Have all data underlying the figures and results presented in the manuscript been provided?**

Reviewer #1: Yes

Reviewer #2: Yes

PLOS authors have the option to publish the peer review history of their article (what does this mean?). If published, this will include your full peer review and any attached files.

Reviewer #1: **Yes: **Brian S Clark

Reviewer #2: No

---

## [Editor Report · Decision Letter 2]

7 Aug 2023

Dear Dr Mumm,

We are pleased to inform you that your manuscript entitled "Transcriptomic comparison of two selective retinal cell ablation paradigms in zebrafish reveals shared and cell-specific regenerative responses" has been editorially accepted for publication in PLOS Genetics. Congratulations!

Yours sincerely,

Gregory S. Barsh

Editor-in-Chief

PLOS Genetics

Gregory Copenhaver

Editor-in-Chief

PLOS Genetics

Comments from the reviewers (if applicable):

**Data Deposition**

http://datadryad.org/submit?journalID=pgenetics&manu=PGENETICS-D-23-00153R2

**Press Queries**

---

## [Editor Report · Acceptance letter]

6 Oct 2023

PGENETICS-D-23-00153R2 

Transcriptomic comparison of two selective retinal cell ablation paradigms in zebrafish reveals shared and cell-specific regenerative responses 

Dear Dr Mumm, 

We are pleased to inform you that your manuscript entitled "Transcriptomic comparison of two selective retinal cell ablation paradigms in zebrafish reveals shared and cell-specific regenerative responses" has been formally accepted for publication in PLOS Genetics! Your manuscript is now with our production department and you will be notified of the publication date in due course.

With kind regards,

Zsofi Zombor

PLOS Genetics

On behalf of:
